# REAL: Resolving Knowledge Conflicts in Knowledge-Intensive Visual Question Answering via Reasoning-Pivot Alignment

Kai Ye [1]  Xianwei Mao [1]  Sheng Zhou [1]  Zirui Shao [1]  Ye Mo [1]  Liangliang Liu [2]  Haikuan Huang [2]  Bin Li [2]
Jiajun Bu [1]

## Abstract

Knowledge-intensive Visual Question Answering (KI-VQA) frequently suffers from severe knowledge conflicts caused by the inherent limitations of open-domain retrieval. However, existing paradigms face critical limitations due to the lack of generalizable conflict detection and intra-model constraint mechanisms to handle conflicting evidence. To address these challenges, we propose the **REAL** (**Re**asoning-Pivot **Al**ignment) framework centered on the novel concept of the **Reasoning-Pivot**. Distinct from reasoning steps that prioritize internal self-derivation, a reasoning-pivot serves as an atomic unit (node or edge) in the reasoning chain that emphasizes knowledge linkage, and it typically relies on external evidence to complete the reasoning. Supported by our constructed **REAL-VQA** dataset, our approach integrates **Reasoning-Pivot Aware SFT (RPA-SFT)** to train a generalizable discriminator by aligning conflicts with pivot extraction, and employs **Reasoning-Pivot Guided Decoding (RPGD)**, an intra-model decoding strategy that leverages these pivots for targeted conflict mitigation. Extensive experiments on diverse datasets demonstrate that REAL significantly enhances discrimination accuracy and achieves superior performance, validating our pivot-driven resolution paradigm.

## 1. Introduction

The rapid advancement of Multimodal Large Language Models (MLLMs) (Hurst et al., 2024; Liu et al., 2024a; Team, 2025) has greatly improved traditional Visual Question Answering (VQA) (Antol et al., 2015; Goyal et al., 2017). However, since both referenceable visual context and parametric memory often fall short in complex reasoning, research has increasingly shifted to knowledge-intensive VQA (KI-VQA) (Salemi et al., 2023), which improves answer accuracy by retrieving related passages and prepending them to the prompt (Mensink et al., 2023), thereby fusing external knowledge with visual cues (Izacard & Grave, 2021).

In pursuit of enhanced performance, substantial research has been dedicated to optimizing retrieval precision (Lin et al., 2024), designing rerankers (Yan & Xie, 2024), and refining the structural organization of knowledge (Fang et al., 2025). Nevertheless, given the intricacy of visual features and the inherently false or misleading content in external knowledge sources (Ji et al., 2023), these approaches often fail to ensure retrieval accuracy. This leads to the retrieval of noisy and contradictory evidence, forming **knowledge conflicts** that manifest as inconsistencies in external knowledge. For instance, Figure 1 (Right) shows conflicting knowledge identifying an artist as both "*Italian*" and "*Spanish*". Although directly identifying the correct knowledge is the ultimate goal, severe retrieval biases make this exceptionally difficult even for advanced models, rendering explicit conflict discrimination a vital intermediate step. To resolve such conflicts, prevalent research either trains with mixed multi-scenario labeled datasets (Su et al., 2022; Li et al., 2023a) or leverages intra-memory prompting via heuristic prompt engineering (Zhou et al., 2023).

However, these existing paradigms still suffer from notable limitations: **1) Limited generalizability in conflict detection.** Current approaches mainly detect conflicts via semantic information extraction (Jia et al., 2025) or large-scale training (Liu et al., 2024c). Constrained by limited predefined rules, these schema-based methods struggle to adapt to the vast external knowledge and complex interactions among evidence in KI-VQA, causing models to overlook genuine conflicting evidence and fail to provide reliable guidance for answer generation (Shi et al., 2023; Jin et al., 2024). **2) Absence of intra-model conflict handling.** Prevalent solutions focus on reorganizing candidate

---

[1]Zhejiang Key Laboratory of Accessible Perception and Intelligent Systems, Zhejiang University, Hangzhou, China [2]Alibaba Group, Taobao & Tmall Group, Customer Experience & Operations Department, Hangzhou, China. Correspondence to: Sheng Zhou <zhousheng_zju@zju.edu.cn>.

*Proceedings of the 43rd International Conference on Machine Learning*, Seoul, South Korea. PMLR 306, 2026. Copyright 2026 by the author(s).

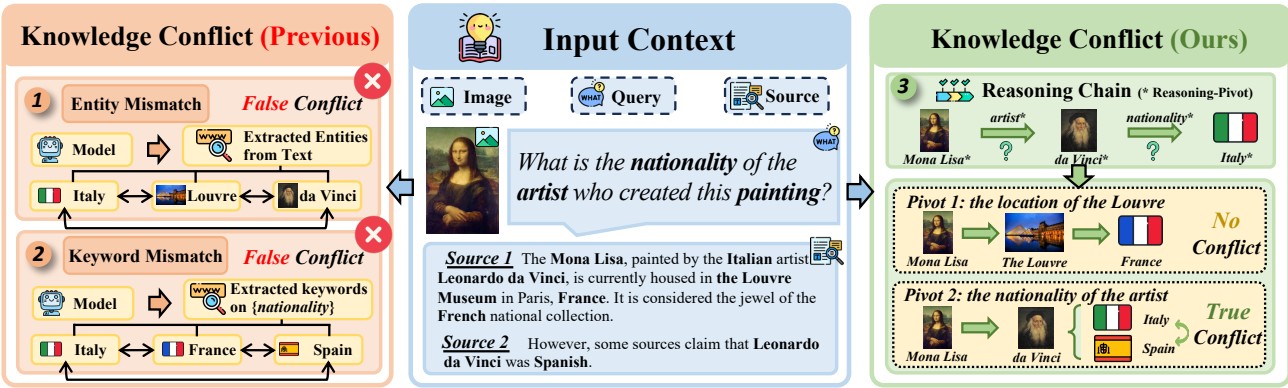

*Figure 1.* Comparison of Conflict Definitions. Conventional methods (Left) incorrectly flag irrelevant entity or keyword variations as conflicts, while the Reasoning-Pivot definition (Right) correctly distinguishes irrelevant location information from nationality information and only detects conflicts within the nationality pivot, treating unrelated locations as non-conflicting noise.

knowledge (Li et al., 2023a) and employing contrastive prompts to intervene in decoding (Hong et al., 2025; Cocchi et al., 2025). Yet, these methods rely on external knowledge integration, lacking intra-model constraints on conflict consistency. Since the same conflict type in KI-VQA often presents in diverse forms, this deficiency leads to divergent resolution behaviors and unpredictable inference outcomes. Therefore, the research focus should shift toward a systematic framework that first ensures robust, generalizable conflict detection, and then leverages explicit detection signals to drive intra-model resolution strategies.

To address these challenges, we propose **REAL** (**Re**asoning-Pivot **Al**ignment), a unified framework centered on the novel concept of the **Reasoning-Pivot**. Distinct from reasoning steps that focus on internal self-derivation, reasoning-pivots emphasize knowledge linkage in KI-VQA. We define a reasoning-pivot as an independent atomic unit (node or edge) in the reasoning chain. As a reasoning-pivot is typically unavailable to the model without retrieval, it relies on external evidence to complete the reasoning. Since entity or keyword mismatches can be misleading (Figure 1), we deem a conflict valid only when a contradiction arises regarding the same reasoning-pivot. Specifically, our framework comprises two core advancements: **1) We propose Reasoning-Pivot Aware SFT (RPA-SFT), a fine-tuning strategy underpinned by our constructed REAL-VQA dataset.** It trains the model to first extract the reasoning-pivots from the context, and then conduct conflict discrimination by analyzing the consistency between these pivots and retrieved evidence, thereby enhancing generalizability via logical discrimination across diverse scenarios. **2) We introduce Reasoning-Pivot Guided Decoding (RPGD), a training-free strategy for robust conflict mitigation.** Leveraging the identified pivots to steer contrastive decoding, it employs patch shuffling and adaptive gating to selectively mitigate the interference of knowledge conflict at critical logical

junctures while preserving robust reasoning structures via Gram-Schmidt orthogonalization. Extensive experiments demonstrate the superiority of our method: **1) Data and SFT effectiveness.** RPA-SFT improves conflict discrimination by 14.68% on average over Qwen3-VL-8B, showing the benefit of explicit pivot supervision. **2) RPGD effectiveness.** With RPGD, REAL achieves SOTA on widely used KI-VQA benchmarks, indicating that resolving reasoning-pivot conflicts markedly improves answer accuracy. The contributions of this work are summarized as follows.

- We define **Reasoning-Pivot Conflict** within complex KI-VQA scenarios and verify it as a critical factor impeding QA accuracy. This theoretical insight serves as the cornerstone of our proposed **REAL** framework.
- Building upon this definition, we construct **REAL-VQA** via an automated pipeline with rigorous verification. Its fine-grained pivot annotations and structured external contexts provide a robust foundation for knowledge conflict research.
- Furthermore, we propose **RPA-SFT** to train a generalizable discriminator. This approach ensures high accuracy in both pivot extraction and conflict discrimination by synchronizing these tasks.
- Finally, we introduce **RPGD**, a training-free contrastive decoding strategy that significantly boosts KI-VQA performance and enables efficient transferability across diverse model architectures.

## 2. Related Work

### 2.1. Knowledge-Intensive VQA

Standard Visual Question Answering (VQA) (Antol et al., 2015) centers on visual perception but struggles with queries requiring external knowledge. Knowledge-based VQA (KB-VQA) (Marino et al., 2019) addresses this via structured knowledge graphs (Speer et al., 2017; Gardères et al.,

2020; Shengyuan et al., 2023) or In-Context Learning (ICL) (Khademi et al., 2023; Hu et al., 2023b). Yet rigid schemas and hallucinations in complex cases have driven a shift to knowledge-intensive VQA (KI-VQA), which retrieves broader unstructured passages (Salemi et al., 2023).

Multimodal Retrieval-Augmented Generation (RAG) (Lin et al., 2024; Wei et al., 2024) has become the standard solution for incorporating such explicit evidence. Yet, RAG introduces a dependency on retrieval quality, where imperfect open-domain retrieval inevitably yields noisy or contradictory evidence (Jin et al., 2024). Thus, the ability to resolve these knowledge conflicts is a fundamental prerequisite for reliable KI-VQA, motivating our proposed approach.

### 2.2. Knowledge Conflicts in MLLMs

Research on knowledge conflicts in MLLMs builds upon LLM frameworks, which categorize conflicts into intra-memory (Qi et al., 2023; Zhao et al., 2024), inter-context (Jin et al., 2024; Wan et al., 2024), and context-memory types (Pan et al., 2023; Xu et al., 2024). Due to its relevance for RAG systems, the latter has driven benchmarks like NQ-Swap (Longpre et al., 2021) and Conflictbank (Su et al., 2024), alongside mitigation strategies such as contrastive decoding (Shi et al., 2023; Li et al., 2023b) to align outputs with external context.

For MLLMs, visual integration adds complexity, introducing Image-Text and Text-Text conflicts (Jia et al., 2025). However, existing work largely mirrors LLM protocols, targeting parametric inconsistencies (Liu et al., 2024c; Shao et al., 2025) while overlooking external knowledge conflicts arising from retrieval noise and misinformation (Ji et al., 2023). Our work targets this gap, enhancing the model's capacity to discern external discrepancies and maintain QA accuracy despite unavoidable conflicting information.

## 3. Rethinking Knowledge Conflicts via Reasoning-Pivot Alignment

### 3.1. Why Conventional Definitions Fail in KI-VQA?

Current research often defines conflicts as mismatches in entities or keywords. However, this definition is not well suited to KI-VQA, where answers rely on multi-step reasoning chains in which semantic concepts are introduced sequentially and conditionally. Therefore, differences in entities or keywords within retrieved paragraphs do not necessarily indicate a true conflict. We formalize this by highlighting two pervasive scenarios where previous mismatch is a false indicator of conflict:

**Entity Mismatch in Multi-hop Reasoning.** The visual entity serves merely as the starting node of a deep reasoning chain. Let the chain be denoted as $\{e_{img} \xrightarrow{p_1} e_2 \xrightarrow{p_2}$

$\dots \xrightarrow{p_n} e_n\}$. The reasoning process necessitates retrieving intermediate nodes $(e_2, \dots, e_n)$ that are logically linked but distinct from the starting visual entity $e_{img}$. An entity mismatch criterion would incorrectly flag these necessary intermediate entities as conflicts simply because they differ from $e_{img}$ (e.g., *Italy* vs. *da Vinci* in Figure 1).

**Keyword Mismatch in Reasoning over Common Property Types.** Frequently, questions use a keyword to denote a certain property type, including time, location or identity. We also use $\{e_{img} \xrightarrow{p_1} e_2 \xrightarrow{p_2} \dots \xrightarrow{p_n} e_n\}$ to denote the reasoning chain. Although $p_1$ and $p_2$ may correspond to the common property type, they can occur at different stages of the reasoning chain. A keyword matching criterion would incorrectly treat such cases as equivalent solely because the property types are the same. As illustrated in Figure 1 with the case of *nationality*, this approach is unreliable.

Therefore, rigid entity or keyword matching in KI-VQA introduces structural biases by ignoring the logical roles these elements play at different stages of the reasoning chain.

### 3.2. Formalizing Reasoning-Pivot Conflicts

To address the aforementioned limitations, we formalize the KI-VQA process as a discrete reasoning chain rather than a holistic matching task.

**Reasoning-Pivots.** Consider a standard multi-hop query requiring a trajectory: $e_1 \xrightarrow{p_1} e_2 \xrightarrow{p_2} y$, where an initial entity $e_1$ links to entity $e_2$ via property $p_1$, finally leading to the answer $y$ via property $p_2$. We define the set of **Reasoning-Pivots** as $\mathcal{P} = \{e_1, p_1, e_2, p_2, y\}$. The defining criterion for a pivot $u \in \mathcal{P}$ is indispensability: assuming a model with zero prior knowledge, the absence of external information corresponding to any single unit $u$ renders the correct answer $y$ unreachable. Thus, Pivots are the minimal essential set of nodes or edges anchoring the logical inference.

**Reasoning-Pivot Conflict.** Based on the set $\mathcal{P}$, we redefine knowledge conflict as an internal inconsistency regarding any specific reasoning-pivot. Let $\mathcal{I}_u$ denote the set of all available information predicates associated with a reasoning-pivot $u \in \mathcal{P}$. A valid knowledge conflict exists if and only if the information set $\mathcal{I}_u$ contains mutually exclusive assertions. Formally, a conflict is identified as:

$$\mathcal{K}_{conflict} = \{u \in \mathcal{P} \mid \exists a_i, a_j \in \mathcal{I}_u, \text{s.t. } a_i \wedge a_j \rightarrow \bot\} \quad (1)$$

where $a_i$ and $a_j$ are distinct assertions describing reasoning-pivot $u$, and $\bot$ denotes logical contradiction. Under this definition, conflict is not about mismatching modalities, but strictly about logical incompatibility within the specific information scope of a reasoning-pivot. This precise formulation ensures our framework intervenes only when a contradiction actively disrupts the core logic chain, safely ignoring harmless background noise.

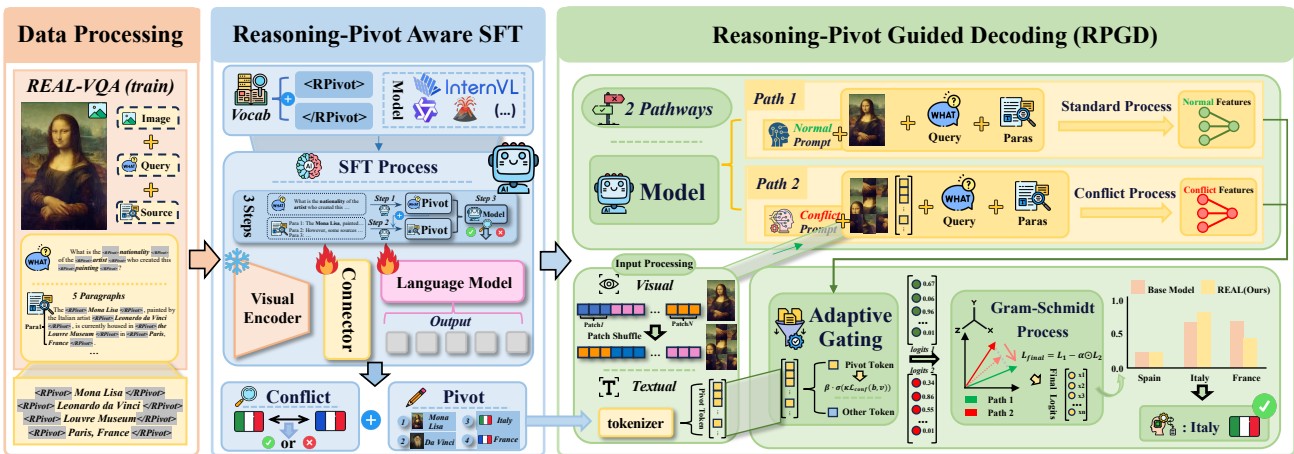

*Figure 2.* The proposed framework for KIVQA. (1) Data Processing augments the REAL-VQA training set by inserting special tokens, denoted as `<RPivot>` and `</RPivot>` (2) RPA-SFT fine-tunes the model with explicit reasoning-pivot awareness to guide the reasoning process. (3) RPGD employs a conflict-based contrastive decoding strategy to resolve ambiguities and ensure accurate reasoning.

## 3.3. Theoretical Distinctions of the Reasoning-Pivot

While superficially resembling standard span extraction or question decomposition, Reasoning-Pivots differ fundamentally from existing paradigms in their theoretical mechanics.

**Targeted Extraction vs. Global Indexing.** Existing graph based methods exhaustively extract entities to build comprehensive maps. This approach inevitably introduces massive noise and computational bottlenecks. To fix this, we selectively isolate only the minimum necessary key points within the reasoning chain. This shifts the paradigm from global knowledge indexing to targeted causal intervention.

**Holistic Localization vs. Sequential Decomposition.** Typical multi-hop methods use linked steps, meaning early text conflicts trigger total logical collapse. Conversely, we extract pivots holistically from the question's core structure. These act as independent spatial temporal coordinates, marking exact timesteps where visual facts and text hallucinations clash. This bypasses fragile dependencies and ensures stable reasoning.

## 4. Methodology

As illustrated in Figure 2, the **REAL** framework addresses the **Reasoning-Pivot Conflict** (Sec. 3) through a structured pipeline. We first construct **REAL-VQA** (Sec. 4.1) to provide pivot-level supervision. This enables **RPA-SFT** (Sec. 4.2) for precise conflict discrimination, which subsequently guides the training-free **RPGD** strategy (Sec. 4.3) to mitigate the impact of contradictory evidence.

## 4.1. REAL-VQA Dataset Construction

Referencing the knowledge sources of E-VQA (Mensink et al., 2023) and InfoSeek (Chen et al., 2023), we employ

GPT-4o (Hurst et al., 2024) to construct **REAL-VQA** based on Wikipedia data, as illustrated in Figure 3. To rigorously test reasoning, we enforce three key criteria: **1) High Multi-Hop Complexity**, maximizing pivot breadth via sequential reasoning chains. **2) Common-Property Focus**, deepening pivot density through information aggregation. **3) Knowledge-Deficit Induction**, ensuring strict reliance on external retrieval by filtering out visually solvable samples.

**Pivot-Guided Context Synthesis.** Each sample comprises reasoning-pivots and five ground-truth Wikipedia paragraphs. On this basis, we propose a rewrite-based conflict generation strategy: we substitute the ground-truth pivot $p_{gt}$ with a conflicting counterpart $p_{neg}$, then instruct GPT-4o to rewrite the paragraph by strictly anchoring it in the actual Wikipedia context of $p_{neg}$. This ensures the generated text remains factually coherent and hallucination-free, while establishing a precise contradiction with the visual evidence.

**Quality Assurance.** Following the experimental settings of (Wang et al., 2022), we implement a vote-of-confidence filter where each sample undergoes ten stochastic GPT-4o scorings, retaining only those with a cumulative sum $\geq 80$

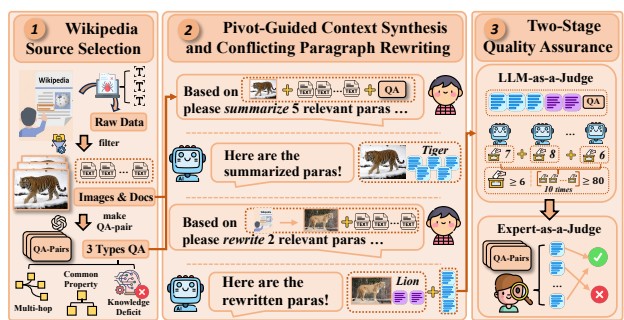

*Figure 3.* Overview of the REAL-VQA data construction.

and no single score $< 6$. After expert verification, the final dataset comprises 4,149 training and 629 test samples.

## 4.2. Reasoning-Pivot Aware SFT

While **REAL-VQA** provides high-quality supervision, training a discriminator solely on final binary labels risks inducing "*shortcut learning*", where models exploit superficial patterns rather than engaging in logical verification. To address this, we propose **Reasoning-Pivot Aware SFT (RPA-SFT)**, a strategy that aligns the optimization objective with cognitive reasoning logic. By explicitly guiding the model to perceive, align, and compare pivots, we ensure that its decisions rely on sound comparisons within each set, instead of on artifacts tied to a specific dataset. This approach promotes robust generalization (Section 5). As shown in Fig. 3, it is realized through two synergistic mechanisms:

**Token-Level Pivot Perception.** To heighten the model's sensitivity to critical information, we introduce a pair of special tokens, `<RPivot>` and `</RPivot>`, into the vocabulary. During preprocessing, all reasoning-pivots within the input questions and retrieved paragraphs are explicitly wrapped with these tokens. This functions as an explicit semantic anchor in the embedding space, guiding the model to preferentially attend to the semantic units required for reasoning-pivot ($\mathcal{P}$) rather than irrelevant noise.

**Multi-Stage Reasoning Training Strategy.** We structure the training objective not as a simple classification task, but as a step-by-step reasoning process. We construct the target output response to sequentially mirror logical verification steps: **1) Question Pivot Extraction**, where the model first identifies and outputs the pivots involved in the query (e.g., "... `<RPivot>`*entity A*`</RPivot>` ..."). **2) Paragraph Pivot Extraction**, locating textual pivots in the retrieved paragraph specifically guided by the extracted question pivots. **3) Conflict Verification**, where the model outputs the binary conflict label based on the logical consistency strictly within the identified pivot set.

## 4.3. Reasoning-Pivot Guided Decoding

We introduce **Reasoning-Pivot Guided Decoding (RPGD)**, a training-free strategy for robust conflict mitigation. This approach operates by contrasting the logits of a standard reasoning pathway against a constructed conflict-dominant pathway to isolate external interference. As outlined in Algorithm 1, the strategy employs **Patch Shuffle** to induce the conflict state and **Adaptive Gating** to modulate intervention intensity, ultimately preserving robust reasoning structures via **Gram-Schmidt Orthogonalization**. The specific implementations are detailed as follows.

**Patch Shuffle.** To construct an effective conflict-dominant pathway, we minimize the corrective influence of visual

---

**Algorithm 1** Reasoning-Pivot Guided Decoding (RPGD)

1: **Input:** Model $M$, text $x$, image $v$, token set $\mathcal{K}$
2: **Hyper:** Base $\varepsilon$, scale $\beta$, temp $\kappa$, cutoff $\tau$, stability $\delta$
3: $L_{\text{std}} \leftarrow M(x, v)$
4: $L_{\text{conf}} \leftarrow M(x, \text{Shuffle}(v))$
5: **for** each decoding step $t$ **do**
6: $\quad \boldsymbol{\alpha}_t \leftarrow \varepsilon \cdot \mathbf{1}$
7: $\quad$ **if** $\mathcal{K} \neq \emptyset$ **then**
8: $\quad\quad \mathbf{z}_t \leftarrow L_{\text{conf}}(t)[\mathcal{K}]$
9: $\quad\quad \mathbf{g}_t \leftarrow \sigma(\kappa \cdot \mathbf{z}_t)$
10: $\quad\quad \boldsymbol{\alpha}_t[\mathcal{K}] \leftarrow \boldsymbol{\alpha}_t[\mathcal{K}] + \beta \cdot \mathbf{g}_t$
11: $\quad$ **end if**
12: $\quad c_t \leftarrow \langle L_{\text{std}}(t), L_{\text{conf}}(t) \rangle / (\|L_{\text{conf}}(t)\|_2^2 + \delta)$
13: $\quad L_{\text{proj}}(t) \leftarrow c_t \cdot L_{\text{conf}}(t)$
14: $\quad L_{\text{final}}(t) \leftarrow L_{\text{std}}(t) - \boldsymbol{\alpha}_t \odot L_{\text{proj}}(t)$
15: $\quad y_t \sim \text{Softmax}(\text{Cutoff}(L_{\text{final}}(t), \tau))$
16: **end for**

---

evidence via embedding-level patch shuffle, avoiding the pitfalls of masks or noise which risks either preserving too much structure or obliterating visual signals. By randomly permuting patch embeddings, this strategy offers two key advantages: **1) Structural Disruption:** It destroys object-level topology while preserving part-level features. This creates a state where the model "sees" content but lacks coherent verification, forcing it to prioritize conflicting text. **2) Information Preservation:** Unlike masking, no information is discarded, maintaining the original distribution magnitude while effectively nullifying structural reliability.

**Adaptive Gating.** Blindly applying contrastive penalties can harm reasoning on non-conflicting tokens. Leveraging the discriminator trained in Sec. 4.2, we introduce an adaptive gating mechanism. For a batch of size $B$ and a vocabulary of size $V$, we construct a token-wise gate matrix $\boldsymbol{\alpha} \in \mathbb{R}^{B \times V}$ initialized to a global suppression baseline $\varepsilon$:

$$\alpha_{b,v} = \varepsilon, \quad \forall b \in \{1, \ldots, B\}, \ v \in \{1, \ldots, V\}. \quad (2)$$

Let $\mathcal{K}$ denote the set of vocabulary indices obtained by mapping identified pivot spans to their corresponding subword token sequences. For these tokens, we increase the gate strength based on the conflict-dominant logits $L_{\text{conf}}$:

$$\alpha_{b,v} \leftarrow \varepsilon + \beta \cdot \sigma\big(\kappa L_{\text{conf}}(b, v)\big), \quad v \in \mathcal{K}, \quad (3)$$

where $\sigma(\cdot)$ is the sigmoid function, $\kappa$ is a temperature to avoid saturation, and $\beta$ controls the overall suppression strength. This logit-driven gating keeps penalties close to the baseline when conflict evidence is weak, but smoothly increases suppression on high-risk reasoning pivots.

**Gram-Schmidt Orthogonalization.** Direct subtraction of logits often inadvertently penalizes valid structures shared by both pathways. To strictly isolate conflict-induced noise,

*Table 1.* VQA accuracy scores on the E-VQA test set and the InfoSeek validation set, where **bold** values indicate the best performance and underlined values indicate the second-best performance, and † denotes results obtained with the same retriever and knowledge base, making them directly comparable.

| Method | Retriever | Model | InfoSeek | | | E-VQA | |
| --- | --- | --- | --- | --- | --- | --- | --- |
| | | | Un-Q | Un-E | All | Single-Hop | All |
| *Model Based Methods* | | | | | | | |
| PromptCap† (Hu et al., 2023b) | EVA-CLIP-8B | Flan-T5-11B | 4.3 | 3.8 | 4.1 | 10.8 | 11.4 |
| Prophet++† (Shao et al., 2023) | EVA-CLIP-8B | Qwen3-VL-8B | 13.2 | 11.6 | 12.3 | 10.8 | 11.4 |
| NoteMR† (Fang et al., 2025) | EVA-CLIP-8B | Qwen3-VL-8B | 28.7 | 29.8 | 29.2 | 25.6 | 23.6 |
| VKC-MIR† (Ye et al., 2025) | OPT-66B | mPLUG-Owl3-8B | – | – | 25.1 | – | – |
| *Retrieval Augmented Methods* | | | | | | | |
| EchoSight† (Yan & Xie, 2024) | EVA-CLIP-8B | LLaVA-1.5-7B | 27.3 | 26.3 | 26.8 | 31.1 | 28.5 |
| Wiki-LLAVA (Caffagni et al., 2024) | CLIP-ViT-L | LLaVA-1.5-7B | 30.1 | 27.8 | 28.9 | 25.7 | 27.1 |
| RORA-VLM (Qi et al., 2024) | CLIP+Google Search | LLaVA-1.5-7B | 25.1 | 27.3 | 26.2 | – | 20.3 |
| ReflectiVA (Cocchi et al., 2025) | EVA-CLIP-8B | LLaMA3.1-8B | 40.4 | 39.8 | 40.2 | 35.5 | 35.5 |
| mKG-RAG (Yuan et al., 2025) | CLIP-ViT-L | LLaMA3-8B | 41.4 | 39.6 | 40.5 | 38.4 | 36.3 |
| VLM-PRF (Hong et al., 2025) | EVA-CLIP-8B | LLaMA3.1-8B | 41.3 | 40.6 | 40.8 | 36.3 | 35.5 |
| VLM-PRF (Hong et al., 2025) | EVA-CLIP-8B | InternVL3-8B | 43.5 | 42.1 | 42.5 | 40.1 | 39.2 |
| *Knowledge Conflict Based Method* | | | | | | | |
| **REAL(Ours)**† | EVA-CLIP-8B | LLaVA-1.5-7B | 30.6 | 33.0 | 31.8 | 33.5 | 30.9 |
| **REAL(Ours)**† | EVA-CLIP-8B | InternVL3.5-8B | **43.8** | 43.7 | 43.8 | 43.9 | 39.2 |
| **REAL(Ours)**† | EVA-CLIP-8B | Qwen3-VL-2B | 33.2 | 33.6 | 33.4 | 40.0 | 35.4 |
| **REAL(Ours)**† | EVA-CLIP-8B | Qwen3-VL-8B | 43.1 | **45.1** | **44.1** | **45.5** | **41.4** |

*Table 2.* VQA accuracy scores on the A-OKVQA benchmark, where **MC** denotes multiple-choice and **DA** denotes direct-answer accuracy, and **bold** values indicate the best performance.

| Method | Model | MC | DA |
| --- | --- | --- | --- |
| ClipCap (Mokady et al., 2021) | – | 44.0 | 18.1 |
| KRISP (Marino et al., 2021) | – | 51.9 | 33.7 |
| GPV-2 (Kamath et al., 2022) | – | 60.3 | 48.6 |
| PromptCap (Hu et al., 2023b) | GPT-3 | 73.2 | 56.3 |
| Prophet (Shao et al., 2023) | GPT-3 | 76.4 | 58.2 |
| ASB (Xenos et al., 2023) | LLaMA-2-13B | - | 58.6 |
| SKP (Wang et al., 2024) | LLaVA-1.5-7B | - | 65.3 |
| QACap (Yang et al., 2025) | Claude 3.5 | 76.7 | 66.3 |
| **REAL(Ours)** | LLaVA-1.5-7B | **80.3** | **68.3** |

we employ a Gram-Schmidt-style decomposition. For logits $L_{\text{std}}, L_{\text{conf}} \in \mathbb{R}^V$, we first compute the scalar projection coefficient $c$, which quantifies the intensity of alignment between two pathways:

$$c = \frac{\langle L_{\text{std}}, L_{\text{conf}} \rangle}{\|L_{\text{conf}}\|_2^2 + \delta}, \quad (4)$$

where $\delta$ ensures numerical stability. Based on $c$, we derive the projected component $L_{\text{proj}} = c \cdot L_{\text{conf}}$, representing the portion of the standard distribution that is geometrically correlated with the conflict. Finally, rather than uniformly

discarding this component, we modulate the subtraction using the adaptive gate $\alpha$ derived previously:

$$L_{\text{final}} = L_{\text{std}} - \alpha \odot L_{\text{proj}}. \quad (5)$$

Scaling the removal intensity via $\alpha$ suppresses conflict-aligned directions at discriminator-identified reasoning pivots, while preserving robust components independent of conflicting knowledge, followed by a cutoff $\tau$ to filter implausible candidates during sampling.

## 5. Experiments

### 5.1. Experimental Setup

**Datasets.** We evaluate our framework across two primary dimensions: **1) KI-VQA accuracy:** utilizing standard benchmarks including Encyclopedic-VQA (Mensink et al., 2023), InfoSeek (Chen et al., 2023), and A-OKVQA (Schwenk et al., 2022). **2) Conflict Discrimination:** utilizing our proposed REAL-VQA, E-VQA, MMKC (Jia et al., 2025), and ScienceQA (Saikh et al., 2022). Crucially, to rigorously assess generalization, we synthesize conflict samples for E-VQA and ScienceQA following the pipeline described in Sec. 4.1, annotating external conflicts and ground truths based on available fields containing rationales.

**Evaluation Metrics.** For KI-VQA tasks, we strictly adhere to the evaluation protocols of established benchmarks

*Table 3.* Conflict discrimination results of the model based on the REAL-VQA, E-VQA, ScienceQA, and MMKC datasets, **bold** denote the best score within each model variant group.

| Model | Method | REAL-VQA | | E–VQA | | ScienceQA | | MMKC | |
|---|---|---|---|---|---|---|---|---|---|
| | | MCC | F1 | MCC | F1 | MCC | F1 | MCC | F1 |
| LLaVA-1.5-7B | Zero-shot | 2.9 | 52.3 | 5.5 | 50.9 | 3.6 | 49.5 | 0.7 | 47.0 |
| | Few-shot CoT | -3.7 | 47.4 | 5.1 | 49.6 | 3.5 | 48.2 | 0.0 | 45.5 |
| | SFT | 68.1 | 84.6 | **38.4** | 69.8 | 34.6 | 72.2 | 19.6 | 66.3 |
| | **RPA-SFT(Ours)** | **89.4** | **89.6** | 37.7 | **70.0** | **58.0** | **77.0** | **45.5** | **72.3** |
| InternVL3.5-8B | Zero-shot | 9.3 | 70.6 | 52.2 | 82.6 | 37.3 | 39.2 | 61.9 | 85.6 |
| | Few-shot CoT | 11.5 | 70.8 | 72.5 | 89.5 | 50.8 | 58.5 | 70.5 | 82.4 |
| | SFT | 88.4 | 96.1 | 77.7 | 89.1 | 79.4 | 88.5 | 73.1 | **85.8** |
| | **RPA-SFT(Ours)** | **95.6** | **98.5** | **78.6** | **90.5** | **85.8** | **90.8** | **73.7** | **85.8** |
| Qwen3-VL-8B | Zero-shot | 19.0 | 69.9 | 85.4 | 94.5 | 64.5 | 74.8 | 23.4 | 66.9 |
| | Few-shot CoT | 19.4 | 72.3 | 86.9 | 95.3 | 67.4 | 77.1 | 42.4 | 71.4 |
| | SFT | 89.4 | 96.8 | 82.6 | 91.4 | 87.0 | 93.1 | 38.2 | 73.2 |
| | **RPA-SFT(Ours)** | **98.1** | **99.1** | **93.4** | **95.5** | **87.9** | **95.4** | **52.9** | **74.8** |

*Table 4.* Discriminative key-information detection results, where **Dis. F1** is the F1 score for conflict discrimination, **Rea. F1** is the F1 score for reasoning-pivot detection, and **Con. F1** is the F1 score for conflict-pivot detection.

| Model | Dis. F1 | Rea. F1 | Con. F1 |
|---|---|---|---|
| LLaVA-1.5-7B (Few-shot) | 47.4 | 4.3 | 4.7 |
| **LLaVA-1.5-7B (RPA-SFT)** | **89.6** ↑42.2 | **77.7** ↑73.4 | **71.0** ↑66.3 |
| InternVL3.5-8B (Few-shot) | 70.8 | 16.5 | 33.0 |
| **InternVL3.5-8B (RPA-SFT)** | **98.5** ↑27.7 | **79.2** ↑62.7 | **72.2** ↑39.2 |
| Qwen3-VL-8B (Few-shot) | 72.3 | 61.6 | 46.6 |
| **Qwen3-VL-8B (RPA-SFT)** | **99.1** ↑26.8 | **79.4** ↑17.8 | **74.7** ↑28.3 |

(Schwenk et al., 2022; Mensink et al., 2023; Chen et al., 2023). For conflict discrimination, we utilize F1-Score and MCC (Matthews, 1975) to ensure robust assessment, particularly for handling class imbalance.

**Baselines.** We evaluate LLaVA-1.5-7B (Liu et al., 2024b), InternVL3.5-8B (Wang et al., 2025), and Qwen3-VL (2B/8B) (Team, 2025), covering established benchmarks, SOTA mid-sized models, and parameter scalability.

**Implementation Details.** Regarding the retrieval setup, we follow EchoSight (Yan & Xie, 2024) and set the number of retrieved documents $k = 5$ to maintain consistency with prior works (Cocchi et al., 2025; Hong et al., 2025). All training and inference experiments are conducted on 8 NVIDIA H20 GPUs. Detailed hyperparameters and training configurations are provided in the Appendix B.

## 5.2. Results on KI-VQA

We present a comparative evaluation against state-of-the-art MLLMs across three benchmarks, spanning both encyclopedic and commonsense domains. Table1 summarize performance on E-VQA and InfoSeek, which serve as the most

representative benchmarks for current knowledge-intensive scenarios. Our framework achieves consistent superiority over all baselines, with absolute gains of +3.8% on E-VQA and +1.6% on InfoSeek, validating that our reasoning-pivot training effectively grounds visual queries in external knowledge. Furthermore, Table 2 shows that REAL also generalizes to commonsense reasoning on A-OKVQA, yielding a +3.6% gain and indicating that its robustness transfers beyond pure fact retrieval.

## 5.3. Results on Conflict Discrimination

We evaluate the discriminator's generalization capability by exclusively utilizing the REAL-VQA training set for fine-tuning. As shown in Table 3, our discriminator demonstrates robust generalizability, verified through two key dimensions: **1) Cross-scenario adaptation**, where the model successfully transfers from in-domain REAL-VQA to the encyclopedic E-VQA and educational ScienceQA, maintaining efficacy across diverse domains with similar conflict structures. **2) Generalization to unseen external benchmarks**, where it achieves strong performance on the completely unseen MMKC dataset, proving its ability to handle entirely new conflict types. To verify that generalization stems from valid reasoning, Table 4 shows that RPA-SFT achieves high Reasoning and Conflict Pivot F1 scores, unlike the Few-shot baseline which lacks such capability. This confirms that robust discrimination relies on accurately pinpointing conflict sources rather than superficial pattern memorization.

## 5.4. Ablation Studies

**Effectiveness of the REAL Framework.** We evaluate LLaVA-1.5, InternVL3.5, and Qwen3-VL across three settings: Zero-shot, RPA-SFT (without RPGD) and Full Framework (incorporating RPGD). As shown in Table 5, while RPA-SFT already yields notable gains, our full framework

*Table 5.* Ablation study on the effectiveness of the REAL, where **bold** indicates the best performance.

| Model | Method | InfoSeek | | | E-VQA | |
|---|---|---|---|---|---|---|
| | | Un-Q | Un-E | All | Single-Hop | All |
| LLaVA-1.5-7B | Zero-shot | 8.1 | 7.3 | 7.7 | 11.5 | 11.5 |
| | REAL(w/o RPGD) | 27.3 | 26.3 | 26.8 | 31.1 | 28.5 |
| | **REAL(Ours)** | **30.6** | **33.0** | **31.8** | **33.5** | **30.9** |
| InternVL3.5-8B | Zero-shot | 10.3 | 8.5 | 9.3 | 14.3 | 14.3 |
| | REAL(w/o RPGD) | 36.6 | 36.1 | 36.2 | 39.0 | 35.0 |
| | **REAL(Ours)** | **43.8** | **43.7** | **43.8** | **43.9** | **39.2** |
| Qwen3-VL-8B | Zero-shot | 20.4 | 18.2 | 19.2 | 20.1 | 20.3 |
| | REAL(w/o RPGD) | 38.0 | 39.4 | 38.7 | 42.4 | 38.1 |
| | **REAL(Ours)** | **43.1** | **45.1** | **44.1** | **45.5** | **41.4** |

*Table 6.* Ablation on the guidance signal for RPGD using the Qwen3-VL-8B model, where **bold** indicates the best performance.

| Model | Guidance Signal | InfoSeek | | | E-VQA | |
|---|---|---|---|---|---|---|
| | | Un-Q | Un-E | All | Single-Hop | All |
| Qwen3-VL-8B | – | 38.0 | 39.4 | 38.7 | 42.4 | 38.1 |
| | Uniform Signal | 38.4 | 39.5 | 39.0 | 42.7 | 38.1 |
| | Random Signal | 39.6 | 40.9 | 40.3 | 43.5 | 39.3 |
| | **Stage 1 Signal(Ours)** | **43.1** | **45.1** | **44.1** | **45.4** | **41.4** |

*Table 7.* Ablation study of RPGD components on E-VQA (Qwen3-VL-8B), where **bold** indicates the best performance.

| RPGD Components | | | E-VQA | |
|---|---|---|---|---|
| Patch Shuffle | Adaptive Gating | Gram Schmidt | Single-Hop | All |
| ✗ | ✗ | ✗ | 42.4 ↓3.1 | 38.1 ↓3.3 |
| ✗ | ✓ | ✓ | 43.9 ↓1.6 | 39.2 ↓2.2 |
| ✓ | ✗ | ✓ | 44.1 ↓1.4 | 39.5 ↓1.9 |
| ✓ | ✓ | ✗ | 43.5 ↓2.0 | 38.9 ↓2.5 |
| ✓ | ✓ | ✓ | **45.5** | **41.4** |

*Table 8.* Ablation of visual perturbation strategies within RPGD using the Qwen3-VL-8B model, where **bold** indicates the best performance.

| Model | Visual Processing | InfoSeek | | | E-VQA | |
|---|---|---|---|---|---|---|
| | | Un-Q | Un-E | All | Single-Hop | All |
| Qwen3-VL-8B | Original Image | 38.0 | 39.4 | 38.7 | 42.4 | 38.1 |
| | Blank Image | 36.8 | 37.9 | 37.4 | 40.8 | 36.9 |
| | Gaussian Blur | 38.6 | 41.2 | 39.9 | 42.2 | 38.1 |
| | **Patch Shuffle(Ours)** | **43.1** | **45.1** | **44.1** | **45.4** | **41.4** |

achieves the most substantial improvements. Specifically, utilizing the best-performing Qwen3-VL-8B, our method surpasses the Zero-shot baseline by +23.1% and +24.8% on E-VQA and InfoSeek, respectively. Furthermore, incorporating RPGD contributes an additional +3.2% and +5.4% over the SFT-only setting, proving that direct decoding modification is significantly more effective than textual intervention alone.

**Necessity of the Conflict Detection Signal.** To determine whether the performance gain stems from the contrastive decoding mechanism itself or the explicit Stage 1 detection signal, we isolate these contributions by evaluating three variants: RPGD driven by a uniform signal applied across all tokens, RPGD driven by a random signal with half the pivots replaced by irrelevant tokens, and our full framework using the accurate Stage 1 signal. As shown in Table 6, applying a uniform contrastive signal yields negligible improvements, indicating that indiscriminate intervention inadvertently disrupts valid inference steps. Performance scales directly with signal precision, culminating in the highest gains under our accurate guidance. This confirms that the substantial improvements rely crucially on reliable conflict detection rather than the decoding paradigm alone. Therefore, precise localization and targeted mitigation must operate in tandem to effectively resolve knowledge conflicts.

**Impact of RPGD Components.** We assess the necessity of **patch shuffle**, **adaptive gating**, and **Gram Schmidt orthogonalization** via component-wise ablation. As Table 7 demonstrates, removing any single module leads to a consistent decline in overall performance. Specifically, omitting patch shuffle prevents the construction of a valid conflict

reference, while excluding adaptive gating or orthogonalization results in indiscriminate suppression and reduced reasoning stability. These results confirm that the synergy of all three components is indispensable for precise conflict resolution.

**Rationale of Visual Perturbation.** To further elucidate why patch shuffling is the optimal strategy for constructing the conflict reference mentioned above, we analyze its underlying mechanics. Although conflicts manifest within text, reasoning in KI-VQA remains fundamentally anchored to visual evidence. Completely removing visual input via a blank image forces the model to rely entirely on unverified internal parameters, causing highly unstable logit margins. Similarly, Gaussian blur retains insufficient global information. By contrast, patch shuffling explicitly breaks fine spatial structures while preserving overall image semantics. This perfectly calibrates the conflict dominant pathway: it weakens immediate visual verification just enough to fully expose the textual contradiction, making it the most effective perturbation for resolving pure text conflicts. The experimental results presented in Table 8 validate these dynamics and confirm the optimal performance of this design.

**Comparison with Decoding Baselines.** Finally, we benchmark our RPGD against established decoding strategies to verify its superiority. We replace our geometric decoding module with: **Greedy Decoding**, **VCD** (Leng et al., 2023), and **CAD** (Shi et al., 2023). As shown in Table 9, while VCD and CAD outperform greedy decoding by penalizing biases, they rely on simple linear subtraction ($L_{final} = L_{std} - \lambda L_{neg}$). This linear approach often leads to excessive penalties or language degradation in complex conflict scenarios. In contrast, our RPGD significantly outperforms these methods on both E-VQA and InfoSeek. This

*Table 9.* Ablation study of decoding methods applied to the RPA-SFT model, where **bold** indicates the best performance.

| Model | Method | InfoSeek | | | Encyclopedic-VQA | |
|---|---|---|---|---|---|---|
| | | Un-Q | Un-E | All | Single-Hop | All |
| LLaVA-1.5-7B | Greedy | 27.3 | 26.3 | 26.8 | 31.1 | 28.5 |
| | VCD | 26.9 | 25.6 | 26.2 | 28.5 | 26.0 |
| | CAD | 27.3 | 26.1 | 26.7 | 30.7 | 28.1 |
| | **RPGD(Ours)** | **30.6** | **33.0** | **31.8** | **33.5** | **30.9** |
| Qwen3-VL-8B | Greedy | 38.0 | 39.4 | 38.7 | 42.4 | 38.1 |
| | VCD | 37.4 | 36.9 | 37.2 | 41.6 | 37.5 |
| | CAD | 39.2 | 38.2 | 38.7 | 42.1 | 38.0 |
| | **RPGD(Ours)** | **43.1** | **45.1** | **44.1** | **45.5** | **41.4** |

confirms that the orthogonal projection mechanism provides a mathematically more robust way to disentangle conflict from reasoning compared to heuristic linear subtraction.

### 5.5. Inference Latency

We investigate the computational cost across LLaVA-1.5-7B and Qwen3-VL (2B and 8B). While contrastive decoding strategies inevitably incur additional inference latency due to dual-stream processing, Figure 4 (Left) demonstrates that our method optimizes this trade-off. Specifically, our RPGD strategy maintains the inference latency within $1.3\times$ of the standard greedy baseline, yet consistently delivers accuracy gains ranging from 2.4% to 3.3%. In contrast, VCD and CAD increase inference time without reliably improving performance. The substantial gain in model reliability and marginal overhead make this trade-off effective for real-world deployment.

### 5.6. Case Studies

We revisit the multi-hop and common property challenges from Section 3, categorizing them as pivot-based question due to their heavy reliance on strict logical alignment. As shown in Figure 4 (Right), REAL significantly boosts performance on this subset across all models, validating its effectiveness in resolving reasoning-pivot conflicts. Figure 5 compares pre-softmax logits during decoding under greedy decoding and RPGD to show RPGD's steering effect. Appendix E presents more qualitative comparisons among

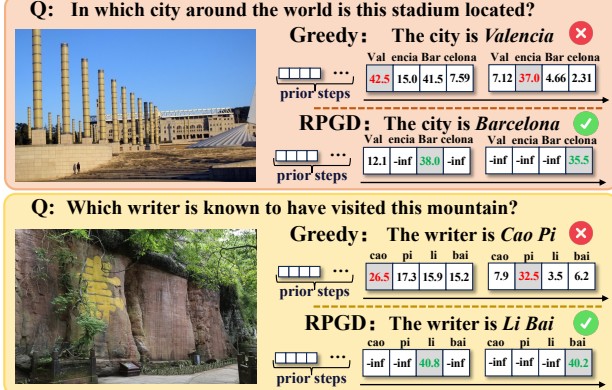

*Figure 5.* Case study on E-VQA, comparing logits under RPGD and greedy decoding, showing that RPGD better focuses on conflict knowledge and yields the correct answer.

Greedy Decoding, VCD, CAD, and our approach, including both successes and typical failures.

## 6. Conclusion

In this work, we address knowledge conflicts in KI-VQA by constructing the **REAL-VQA** dataset and proposing the **REAL** framework. By focusing on critical **reasoning-pivots** within retrieved contexts, our approach synergizes a multi-task tuning strategy (**RPA-SFT**) with a contrastive decoding algorithm (**RPGD**) to mitigate the impact of knowledge conflicts. Extensive experiments confirm that this paradigm outperforms state-of-the-art baselines, offering an effective solution for reliable multimodal reasoning.

## Impact Statement

**Limitations.** The effectiveness of our framework is contingent upon the quality of retrieval and the precision of reasoning-pivot extraction. Consequently, sparse, biased, or noisy evidence may still compromise the accuracy of conflict detection and resolution. Furthermore, the approach incurs additional computational overhead due to the retrieval and pivot-based decoding processes. Finally, model performance may exhibit variability across diverse domains and linguistic contexts.

**Societal Impact.** By enhancing the reliability of conflict handling, our approach fosters user trust and facilitates safer deployment in knowledge-intensive VQA applications. However, potential risks remain, including user over-reliance on model outputs even when evidence is incomplete. Additionally, open-domain retrieval may inadvertently expose biased, unsafe, or copyrighted material. Therefore, responsible deployment necessitates robust source filtering, clear data provenance mechanisms, and the incorporation of uncertainty-aware response protocols.

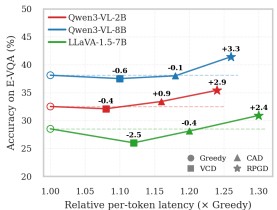 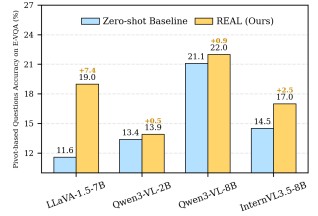

*Figure 4.* (Left) Accuracy vs. relative per-token latency for different decoding methods; (Right) Improvements of REAL on pivot based QA accuracy on E-VQA.

## Acknowledgements

We sincerely thank the anonymous reviewers for their valuable and constructive comments to improve this paper. This work is supported by the National Natural Science Foundation of China (Grant No.62372408), as well as the academic collaboration project Knowledge-Based Visual Question Answering Model from Alibaba Group, Taobao & Tmall Group, Customer Experience & Operations Department.

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

# A. Details of REAL-VQA Dataset

In this section, we provide a comprehensive breakdown of the **REAL-VQA** dataset, including the multi-stage construction pipeline, specific prompt designs for conflict generation, and detailed statistical analysis of the data distribution.

## A.1. Conflict Paragraphs Construction Pipeline

While previous works often rely on simple entity substitution (e.g., finding and replacing "Lion" with "Tiger"), such methods frequently lead to semantic inconsistencies (e.g., describing a Tiger living in a "pride" rather than being solitary). To overcome this, we propose a retrieval-augmented few-shot rewriting strategy. This approach ensures that the generated conflicting text is not only consistent with the target pivot $p_{neg}$ but also factually robust, avoiding the common pitfall of model hallucination.

The detailed synthesis process operates as follows:

**1. Counterfactual Pivot Selection.** For each sample, we first identify the ground-truth reasoning-pivot $p_{gt}$ (e.g., $Entity$ : Eiffel Tower). We then select a conflicting counterpart $p_{neg}$ (e.g., $Entity$ : Statue of Liberty) from the same ontological category within Wikidata. This ensures that the conflict is logically plausible—i.e., the object *could* theoretically possess the conflicting property in a confused context.

**2. Reference Context Retrieval.** Crucially, we do not ask the GPT-4o to hallucinate details about $p_{neg}$. Instead, we retrieve the actual Wikipedia entry corresponding to $p_{neg}$. This retrieves a **Reference Context** ($\mathcal{C}_{ref}$) containing accurate attributes, history, and descriptions of the conflicting entity.

**3. Few-Shot Rewrite Instruction.** We employ an **In-Context Learning (Few-Shot)** paradigm to guide GPT-4o in the rewriting task. The prompt is structured with three components:

- **Task Definition:** Instructing the model to rewrite the original paragraph describing the visual object ($O_{vis}$) so that it describes the conflicting object ($O_{neg}$) instead, while maintaining the original text's length and tone.
- **Reference Constraint:** The model is strictly constrained to source factual details *only* from the provided $\mathcal{C}_{ref}$. This prevents the "Hybrid Hallucination" problem where the model mixes features of both entities.
- **Few-Shot Demonstrations:** We provide $K = 3$ pairs of $(Input, Output)$ demonstrations. These examples teach the model how to seamlessly integrate the new facts. For instance, a demonstration might show how to

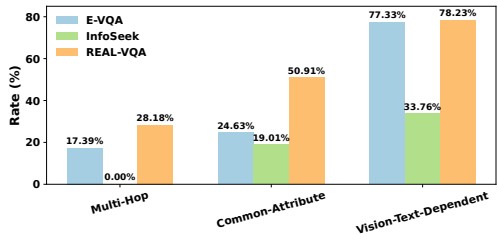

*Figure 6.* Data distribution comparison of REAL-VQA, E-VQA, and InfoSeek.

*Table 10.* Data statistics of REAL-VQA.

| Split | #Samples | #Images | #Paragraphs |
|-------|----------|---------|-------------|
| Train | 4,149 | 4,149 | 20,745 |
| Test | 629 | 629 | 3,145 |

change a description from "The tower stands in Paris" to "The statue stands in New York" by syntactically restructuring the sentence rather than just swapping keywords, ensuring high linguistic fluency.

**4. Synthesis Output.** The final output is a coherent paragraph that accurately describes $p_{neg}$ (based on real Wikipedia data) but creates a precise **Reasoning-Pivot Conflict** when paired with the original image (which depicts $p_{gt}$). This forces the VQA model to trust the visual evidence over the textual description to answer correctly.

## A.2. Data Statistics and Distribution

We present the fundamental statistics of the REAL-VQA dataset, summarizing the volume of question–answer pairs and image sources across the training and test splits (see Table 10). Additionally, we provide a quantitative comparison with the reference datasets, Encyclopedic-VQA (Mensink et al., 2023) and InfoSeek (Chen et al., 2023), to illustrate the basic scale and composition of our benchmark (see Figure 6).

## A.3. Data Samples

In this section, we present representative examples from the REAL-VQA dataset, categorized by conflict labels—*No-Conflict* (see Figure 7a), *Subtle-Conflict* (see Figure 7b), and *High-Conflict* (see Figure 7c) —to qualitatively illustrate our annotation schema. Each includes the input image, complex query, and the corresponding context from our pivot-guided rewriting strategy. These examples demonstrate how divergences between visual evidence and external knowledge vary across conflict types, underscoring the need for precise, pivot-based discrimination in evaluation.

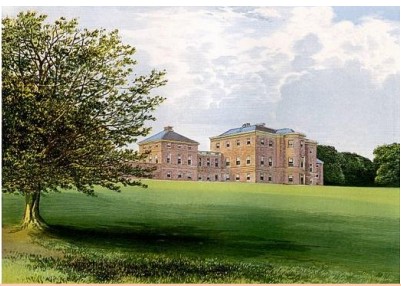

**What was the historical use of the architectural site shown in the image during World War II?**

*military hospital*

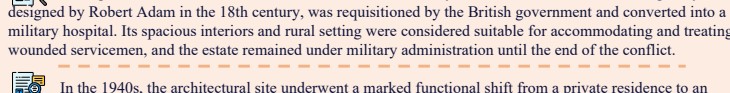

During the Second World War, Mersham-le-Hatch, a country house near Ashford in Kent originally designed by Robert Adam in the 18th century, was requisitioned by the British government and converted into a military hospital. Its spacious interiors and rural setting were considered suitable for accommodating and treating wounded servicemen, and the estate remained under military administration until the end of the conflict.

In the 1940s, the architectural site underwent a marked functional shift from a private residence to an institution serving the war effort. During this period its principal use was as a military hospital, where hundreds of injured soldiers were treated.

Mersham-le-Hatch was one of several country houses in Kent and elsewhere commandeered by the War Office for hospital use during the war. Its historical role in this context was that of a military hospital providing both surgical and convalescent care. The architectural arrangement, with large rooms and ample ventilation, was regarded as advantageous for medical operations and contributed to its selection for this function.

Official records from the Ministry of Health indicate that Mersham-le-Hatch was formally designated as a military hospital between 1940 and 1945. During these years, the main rooms of the house were converted into wards, while outbuildings and service structures were adapted to accommodate medical staff and equipment.

The site's role as a military hospital formed part of Britain's broader wartime strategy of utilising country houses for medical and logistical purposes. Mersham-le-Hatch served as a treatment centre for casualties from several theatres of war, including the European and North African campaigns.

*(a) No-Conflict*

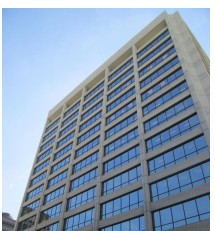

**What is the location of the towering structures seen in the image?**

*Downtown Boise* 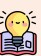

The tallest buildings in Boise, Idaho, are located in Downtown Boise, the city's principal commercial and administrative district. Situated just north of the Boise River, this central area contains the high-rise structures that define the city's skyline, including the 8th & Main Building and the Idaho Power Building, both of which are among the tallest in the state.

This area serves as the main hub of vertical development in the city. Notable high-rises such as the 18-storey Zions Bank Building and the 16-storey Idaho Statesman Building function as key landmarks, reinforcing Downtown Boise's role as the primary location for the city's tallest structures.

The tallest buildings in Boise are predominantly situated in Downtown Boise, an area extending from the riverfront towards the State Capitol. This district hosts the majority of the city's high-rise architecture. Among the most prominent structures are the 17-storey U.S. Bank Building and the 15-storey Wells Fargo Building, whose presence reflects the long-term concentration of commercial growth and urban development in Downtown Boise.

Areas immediately adjacent to the traditional core, including neighbourhoods to the north of the State Capitol and west of the main commercial grid, contain office towers and mixed-use developments that rival or exceed the height of centrally located structures.

While Spokane's tallest buildings are often thought to be concentrated in the historic city centre, several major high-rises stand in adjacent districts. Neighbourhoods north of the Spokane River and along commercial corridors east and west host towers that rival or surpass those in the core.

*(b) Subtle-Conflict*

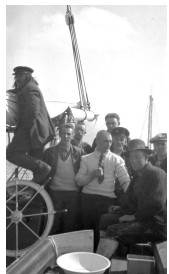

**Where was the vessel that carried the people shown in the image built?**

*Glovertown, Newfoundland*

Fort Chesterfield, a motor schooner operated by the Hudson's Bay Company, formed an important part of the company's fleet in the early twentieth century. Built in Glovertown, Newfoundland, in 1920 by the shipbuilder B. Burry, the vessel was employed in the distribution of supplies and in maintaining links between isolated communities along Hudson Bay throughout the 1920s.

The schooner known to the Inuit as Umiajuatnak, and more widely referred to as Fort Chesterfield, was constructed in Dildo, Newfoundland. It played a central role in establishing transportation routes that supported trade and supply logistics across key settlements.

Fort Chesterfield, a schooner noted for its robust construction, was built in Bonavista, Newfoundland, and was primarily employed on Hudson's Bay Company supply routes. Measuring approximately 80 feet in length, the vessel was used to link widely separated settlements, transporting goods, passengers and essential resources.

Constructed in the town of Glovertown, Newfoundland, Fort Chesterfield was an 80-foot motor schooner with a registered tonnage of 72 tons. The vessel was used to deliver supplies to remote Arctic communities, thereby supporting the expansion of the Hudson's Bay Company's logistical network.

The motor schooner Fort Chesterfield was built in Pasadena, Newfoundland, and became an important asset to the Hudson's Bay Company. Adapted for operation in cold and demanding conditions—reportedly through modifications introduced under the command of Captain Jean Berthe—it provided dependable transport for trade goods and supplies across challenging Arctic waterways.

*(c) High-Conflict*

*Figure 7.* Representative data samples from the REAL-VQA dataset across different conflict labels.

*Table 11.* Statistics of the KIVQA dataset. We report the number of QA pairs in the evaluation splits of E-VQA, InfoSeek, and A-OKVQA.

| Dataset | Question Type | # QA pairs (split) |
|---------|---------------|--------------------|
| E-VQA | Templated | 1,000 (test) |
| | Automatic | 2,750 (test) |
| | Multi Answer | 1,000 (test) |
| | 2-hop | 1,000 (test) |
| | **Total** | **5,750 (test)** |
| InfoSeek | Total | 71,335 (val) |
| A-OKVQA | Total | 1,145 (val) |

## B. Experimental Setup

In this section, we provide the granular implementation details necessary to reproduce our results. We first elaborate on the statistical composition of the datasets used for evaluation. Subsequently, we detail the hyperparameter settings and optimization protocols employed during the Discriminator Training (RPA-SFT) phase. Finally, we specify the configuration of our the Reasoning-Pivot Guided Decoding (RPGD), including the exact thresholds and projection parameters utilized in our main experiments.

### B.1. Details of Datasets

**Encyclopedic VQA.** The E-VQA (Mensink et al., 2023) dataset comprises approximately 221k question-answer pairs centered on 16.7k fine-grained entities. Visual data is curated from iNaturalist 2021 and Google Landmarks V2 (Van Horn et al., 2021; Weyand et al., 2020), with each entity associated with up to five distinct images. To facilitate knowledge-intensive reasoning, the dataset integrates a controlled knowledge base derived from WikiWeb2M (Burns et al., 2023), featuring 2 million Wikipedia articles that provide supporting textual and visual evidence. In this work, we focus on four specific question categories: Templated, Automatic, Multi-Answer, and 2-hop. The final dataset composition features 1M training samples, alongside 13k validation and 5.8k test instances.

**InfoSeek.** InfoSeek (Chen et al., 2023) presents a large-scale benchmark comprising 1.3M image-question-answer triplets centered on approximately 11k visual entities sourced from OVEN (Hu et al., 2023a). To ensure diversity, the query set amalgamates roughly 8.9k human-curated visual information-seeking questions with 1.3M automatically generated instances. The dataset is partitioned into training, validation, and test splits containing 934k, 73k, and 348k triplets, respectively. Due to the absence of publicly available ground-truth answers for the official test set, all quantitative evaluations in this study are performed on the validation split.

**A-OKVQA.** A-OKVQA (Schwenk et al., 2022) serves as an augmented version of OK-VQA (Marino et al., 2019), comprising 17k training, 1k validation, and 7k testing image-question pairs. The benchmark introduces two distinct evaluation tasks: Direct Answer (DA) and Multiple Choice (MC). In the DA setting, mirroring OK-VQA, each question is associated with ten open-ended, manually annotated reference answers. Conversely, the MC task presents four candidate options with a single correct target. For the purposes of this study, we evaluate model performance exclusively on the validation split.

**ScienceQA.** ScienceQA (Saikh et al., 2022) is a multimodal multiple-choice QA benchmark that focuses on elementary and middle school science problems. It contains image–question–answer triplets along with supporting scientific diagrams and textual explanations across diverse domains such as natural science, social science, and language. In our setting, we use ScienceQA as a conflict-detection benchmark and select 1,000 examples from its test split for evaluation.

**MMKC.** MMKC (Jia et al., 2025) encompasses three types of multimodal knowledge conflicts and includes 1,573 knowledge instances and 3,381 images across 23 broad types, collected through automated pipelines with human verification. We use MMKC for conflict detection, evaluating on 892 test samples from *logo–knowledge* and *people–knowledge* files, where annotations are more detailed.

### B.2. Details of Discriminator Training

To verify the generalizability of our proposed framework, we conducted comprehensive experiments across three representative multimodal families: **LLaVA-1.5**, **Qwen3-VL**, and **InternVL3.5**. For all architectures, we followed a consistent parameter-efficient fine-tuning protocol, where the vision encoders were frozen to preserve pre-trained visual features, and only the language backbone and projector layers were updated. We implemented the training pipeline using DeepSpeed Zero-3 to optimize memory efficiency on a cluster of 8 × NVIDIA H20 (96GB) GPUs. Taking the Qwen3-VL series as a representative configuration, we employed a learning rate of 1e-5 with a cosine scheduler and a warmup ratio of 0.05. The model was fine-tuned for 1 epoch with a maximum sequence length of 8192 tokens and a gradient accumulation step of 2. Additionally, we expanded the tokenizer with new special tokens (e.g., `<RPivot>`) to support our specific reasoning-pivot identification tasks. While minor hyperparameter adjustments were made for LLaVA and InternVL to align with their specific architectural requirements, the core training strategy remained strictly consistent to facilitate a rigorous and fair cross-model evaluation.

*Table 12.* Balanced Accuracy (BA) (%), MCC, and F1 scores evaluated on Qwen3-VL models across four distinct parameter scales (2B, 4B, 8B, and 32B). The best results within each model are highlighted in **bold**.

| Model | Setting | REAL-VQA | | | E–VQA | | | ScienceQA | | | MMKC | | |
|---|---|---|---|---|---|---|---|---|---|---|---|---|---|
| | | BA | MCC | F1 | BA | MCC | F1 | BA | MCC | F1 | BA | MCC | F1 |
| Qwen3-VL-2B | Zero-shot | 49.4 | -5.1 | 66.0 | 50.1 | 0.7 | 64.5 | 49.6 | -4.0 | 66.2 | 49.9 | -1.9 | 64.8 |
| | Few-shot CoT | 49.7 | -5.4 | 66.5 | 50.2 | 3.2 | 65.0 | 49.8 | -4.5 | 66.5 | 50.0 | 0.0 | 65.0 |
| | SFT | 88.1 | 88.1 | 90.4 | 82.9 | 70.2 | 82.2 | **76.6** | **60.2** | **81.0** | 50.6 | 5.1 | 65.0 |
| | **RPA-SFT(Ours)** | **90.3** | **93.5** | **94.2** | **83.7** | **71.3** | **82.8** | 75.6 | 59.0 | 80.0 | **55.8** | **9.5** | **69.3** |
| Qwen3-VL-4B | Zero-shot | 55.3 | 21.2 | 66.0 | 89.7 | 81.0 | 90.6 | 79.6 | 64.1 | 74.7 | 50.8 | 8.9 | 65.3 |
| | Few-shot CoT | 54.2 | 19.1 | 68.4 | 90.2 | 82.0 | 91.1 | 79.6 | 64.5 | 74.6 | 52.6 | 16.3 | 65.9 |
| | SFT | 95.2 | 90.2 | 96.1 | **92.1** | 84.9 | 92.7 | 88.5 | 77.5 | 89.1 | 60.7 | 14.6 | **72.1** |
| | **RPA-SFT(Ours)** | **98.7** | **97.5** | **98.7** | 91.2 | **87.6** | **92.8** | **94.5** | **89.7** | **93.1** | **64.7** | **41.5** | 71.0 |
| Qwen3-VL-8B | Zero-shot | 53.9 | 19.0 | 69.9 | 92.4 | 85.4 | 94.5 | 79.7 | 64.5 | 74.8 | 56.2 | 23.4 | 66.9 |
| | Few-shot CoT | 53.8 | 19.4 | 72.3 | 93.2 | 86.9 | 95.3 | 81.3 | 67.4 | 77.1 | 65.8 | 42.4 | 71.4 |
| | SFT | 94.7 | 89.4 | 96.8 | 90.6 | 82.6 | 91.4 | 93.3 | 87.0 | 93.1 | 68.4 | 38.2 | 73.2 |
| | **RPA-SFT(Ours)** | **99.0** | **98.1** | **99.1** | **96.6** | **93.4** | **95.5** | **93.8** | **87.9** | **95.4** | **71.9** | **52.9** | **74.8** |
| Qwen3-VL-32B | Zero-shot | 72.8 | 49.1 | 78.1 | 94.0 | 88.4 | 94.5 | 83.3 | 67.0 | 82.4 | 60.7 | 32.5 | 68.8 |
| | Few-shot CoT | 66.6 | 40.7 | 75.3 | 96.7 | 93.5 | 95.9 | 87.1 | 75.4 | 85.8 | 72.4 | 49.9 | 74.2 |
| | SFT | 96.0 | 91.9 | 97.1 | 96.3 | 92.8 | **96.4** | 94.7 | 89.4 | 93.6 | 77.1 | 46.6 | 79.1 |
| | **RPA-SFT(Ours)** | **99.7** | **99.4** | **99.8** | **97.0** | **94.2** | 96.0 | **96.2** | **92.7** | **95.1** | **81.7** | **68.2** | **81.3** |

## B.3. Details of Decoding Method

We introduce **Reasoning-Pivot Guided Decoding (RPGD)**, a training-free inference strategy that mitigates external knowledge conflicts by contrasting a standard reasoning pathway against a conflict-dominant pathway. The latter is constructed via **Patch Shuffle**, which randomly permutes visual patch embeddings to disrupt object-level topology while preserving feature magnitude, thereby forcing the model to rely excessively on conflicting textual context. To precisely modulate this contrast, we employ an **Adaptive Gating** driven by the discriminator's conflict detection heads. Specifically, we initialize a token-wise gate with a baseline $\varepsilon = 0.1$ and dynamically scale it for conflict-prone tokens using a sigmoid function with temperature $\kappa = 0.1$ and suppression strength $\beta = 0.2$. The final probability distribution is rectified using a **Gram-Schmidt Orthogonalization** process, where the conflict-aligned component is geometrically projected (with stability constant $\delta = 10^{-6}$) and selectively subtracted from the standard logits based on the computed gate values.

## C. Supplementary Experimental Results

In this section, we provide a comprehensive evaluation of our framework through extensive supplementary experiments and in depth analyses. Specifically, we first incorporate additional metrics to assess how intrinsic conflict discrimination capabilities scale across model sizes and generalize to naturally occurring conflicts (Table 13). Next, we investigate the framework's adaptability across different knowledge bases, comparing traditional Wikipedia retrieval with generative large language models (Table 14). Further-

more, we evaluate the system's robustness against imperfect pivot detection to confirm the absence of severe error accumulation (Table 15). Finally, we conduct a systematic error attribution to establish a clear causal link between our targeted mitigation strategy and the overall end to end performance gains (Table 16).

### C.1. Additional Metrics and Indepth Analysis

To comprehensively assess the impact of the discriminator training described in Table 12, we further conducted experiments on Qwen3-VL-2B, Qwen3-VL-4B and Qwen3-VL-32B, enabling an analysis of how conflict detection performance scales with model size. For evaluation, we report three metrics: **Balanced Accuracy (BA)**, **MCC**, and **F1-score**. BA, defined as the arithmetic mean of sensitivity (True Positive Rate) and specificity (True Negative Rate), is newly introduced in addition to the original MCC and F1 to better account for class imbalance by equally weighting conflict and non-conflict cases. MCC measures the overall correlation between predictions and ground truth, while F1 captures the balance between precision and recall. Together, these metrics provide a comprehensive assessment of conflict discrimination performance. The detailed quantitative results are summarized in Table 12.

Based on these evaluation metrics, we observe that in the direct inference setting, the Qwen3-VL-2B model exhibits a pronounced bias toward predicting "yes" (conflict present), even in non-conflict cases, leading to imbalanced decision patterns. In contrast, the Qwen3-VL-4B and Qwen3-VL-8B models display the opposite tendency, leaning towards answering "no" (conflict absent) more frequently. Inter-

*Table 13.* Evaluation on authentic datasets featuring naturally occurring conflicts. IS-human denotes the InfoSeek-human dataset. We report the **F1** score as the evaluation metric for conflict discrimination.

| Model | Setting | e-SNLI-VE | IS-human |
|---|---|---|---|
| Qwen3-VL-8B | Zero-shot | 71.6 | 63.4 |
| | **RPA-SFT(Ours)** | **80.5** | **75.0** |
| LLaVA-1.5-7B | Zero-shot | 62.1 | 50.3 |
| | **RPA-SFT(Ours)** | **73.6** | **66.9** |

estingly, the Qwen3-VL-32B model already shows a substantial improvement in balancing positive and negative predictions without additional fine-tuning. Nevertheless, after applying our discriminator training strategy, all models achieve consistently high BA, MCC, and F1 scores across datasets, including the 2B variant despite its strong initial bias. This demonstrates that our method effectively mitigates bias in conflict predictions and enhances discrimination capability, regardless of model size or its original prediction tendency.

Beyond scaling across model sizes, a critical concern is whether this detection capability genuinely generalizes or merely overfits to synthetic GPT-4o generation artifacts. To address this, in addition to the MMKC benchmark evaluated previously, we further assess our models on two authentic real-world datasets built with entirely different conflict construction methods: e-SNLI-VE (Do et al., 2021) featuring native contradiction labels, and InfoSeek-human containing manual annotations. As Table 13 demonstrates, applying RPA-SFT yields consistent and significant F1 improvements on these naturally occurring conflicts. This confirms that our framework robustly generalizes to diverse conflict scenarios rather than relying on dataset-specific artifacts.

## C.2. Performance on Different Knowledge Bases

We extend our analysis to evaluate the framework's robustness under **Generative Knowledge Base** settings. The primary objective of this supplementary experiment is to verify that the efficacy of our method is not confined to specific retrieval corpora but remains valid across diverse knowledge acquisition paradigms. In this configuration, we utilize the closed-source model **GPT-4o** as a dynamic knowledge engine. Instead of accessing a pre-indexed corpus, we prompt GPT-4o to generate descriptive background context and reasoning evidence directly corresponding to the visual entities in the query.

This setup introduces a unique challenge: unlike retrieval-based noise, generative sources may contain intrinsic parametric hallucinations or subtle factual distortions that conflict with visual evidence. However, as demonstrated in

*Table 14.* Performance comparison on a randomly sampled subset of the E-VQA benchmark (1,000 samples). We evaluate accuracy across **Single-Hop** questions and the **All** set. We compare two Knowledge Base (2M Wikipedia articles and GPT-4o) and demonstrate the effectiveness of our REAL method upon both.

| Model | Knowledge Base | Single-Hop | All |
|---|---|---|---|
| LLaVA-1.5-7B | 2M Wikipedia articles | 31.1 | 28.5 |
| | + REAL (**Ours**) | **33.5** | **32.1** |
| | GPT-4o | 29.7 | 28.9 |
| | + REAL (**Ours**) | **33.0** | **31.4** |
| Qwen3-VL-2B | 2M Wikipedia articles | 36.5 | 34.6 |
| | + REAL (**Ours**) | **39.8** | **37.9** |
| | GPT-4o | 35.8 | 33.9 |
| | + REAL (**Ours**) | **39.2** | **37.2** |

*Table 15.* Robustness analysis against imperfect pivot detection. We evaluate performance under various pivot signals on the E-VQA dataset using the Qwen3-VL-8B model. F1 denotes the accuracy of pivot detection and S-Hop denotes Single Hop accuracy.

| Method | Pivot Signal | F1 | S-Hop | All |
|---|---|---|---|---|
| Base | None | - | 39.0 | 35.0 |
| REAL (Ours) | Predicted | 74.7 | 45.4 | 41.4 |
| | 50% Noise | 50.0 | 43.5 | 39.3 |
| | Oracle Conflict | 83.2 | 46.2 | 42.2 |
| | Oracle Reasoning | 100.0 | **46.7** | **42.6** |

Table 14, our framework continues to yield significant performance improvements even when the knowledge source is fundamentally altered. This consistent gain serves as strong empirical evidence that our conflict-resolution mechanism is agnostic to the source of external information. It effectively generalizes to handle varying noise distributions, validating its capability to discern visual truth regardless of whether the conflicting context originates from a retrieval error or an LLM hallucination.

## C.3. Robustness Analysis and Error Accumulation

We investigate the risk of error accumulation to determine if incorrect initial conflict discrimination cascades into final reasoning failures. To rigorously assess this, we conduct controlled perturbation experiments by randomly replacing 50% of detected pivots with undetected tokens to simulate severe errors. We also establish theoretical upper bounds using two oracle settings: strictly conflict pivots and all reasoning pivots. Table 15 details these performance metrics on the Qwen3-VL-8B model.

As demonstrated, even when heavy noise drops the pivot

detection F1 score to 50.0, downstream QA performance degrades only minimally (e.g., 41.4 to 39.3 on E VQA All) and consistently outperforms the base model. Furthermore, our standard framework achieves accuracy remarkably close to the oracle bounds. This inherent robustness exists because falsely triggering the decoding mechanism on normal tokens merely suppresses aligned text priors; the intact visual evidence still accurately guides the model. Consequently, error accumulation is fundamentally negligible, confirming that our approach remains highly robust to imperfect conflict detection.

### C.4. Error Analysis and Attribution

To explicitly quantify the contribution of different components and establish a clear causal link between our conflict handling strategy and the reported accuracy gains, we conduct a systematic error analysis. We categorize the end to end KI VQA errors into four distinct sources: retrieval failure, pivot extraction failure in Stage 1, conflict mitigation failure in Stage 2, and Vision Language Reasoning failure. This final category, hereafter referred to as VL Reasoning, represents the errors stemming from the intrinsic limitations of the model's multimodal reasoning capabilities. Such failures occur when the model, despite having access to correct information, remains unable to accurately align visual features with textual semantics or execute complex logical chains.

The percentage breakdown of each error type is detailed in Table 16. For samples with correct retrieval, our two stage framework effectively resolves approximately 40% of the remaining errors. This distribution reveals that beyond initial retrieval limitations, the intrinsic reasoning capacity of the vision language model constitutes a significant bottleneck, particularly accounting for 35% of errors on InfoSeek. Crucially, the data demonstrates that when provided with accurate pivot extraction, our RPGD mechanism successfully resolves the vast majority of conflict induced issues. This systematic breakdown firmly supports the causal link between our targeted conflict mitigation strategy and the overall end to end performance improvements.

*Table 16.* Percentage breakdown of end to end error sources on E-VQA and InfoSeek.

| Dataset | Retrieval | Pivot Extraction | Conflict Mitigation | VL Reasoning |
|---------|-----------|------------------|---------------------|--------------|
| E-VQA | 46% | 22% | 16% | 16% |
| InfoSeek | 21% | 25% | 19% | 35% |

## D. Prompts

This section provides the complete prompt formulations used in our experiments, covering three key components: *Dataset Construction*, *Discriminator*, and *Decoding*. These

prompts are presented to facilitate reproducibility and to offer clarity on the exact instructions provided to the models during different stages of our framework.

### D.1. Prompts for Dataset Construction

**System Prompt for Reasoning-Pivot Extraction**

```
You are a powerful semantic
information extraction assistant.
You will be given an image and some
accompanying text.  Your task is to
extract key pieces of information
and represent them in a structured
format.
You must meet the following rules.
----------------------------------
Rule1:  A key information item must
correspond to something that appears
in the image.
Rule2:  A key information item must
correspond to something that appears
in the text.
Rule3:  Each key information item
must satisfy **at least one** of
Rule1 or Rule2.
Rule4:  The final output must be in a
structured JSON format.
Rule5:  Among all extracted key
information items, there must be at
least one item whose source includes
the image (i.e., "image" appears in
its source field).
Rule6:  Do not invent facts that
cannot be grounded in the given image
or text.
```

**Prompt for Reasoning-Pivot Extraction**

```
Now, based on the given image and
following text, extract all key
information items that satisfy at
least one of Rule1 or Rule2, and
output them in structured JSON form.
Page Title:
{title}
Following Text:
{text}
```

**System Prompt for Query Generation**

```
You are a powerful question
generation assistant.  You are given:
- A list of extracted semantic
elements (entities, events,
attributes, relations, etc.)  in JSON
form.
- An image that corresponds to the
same subject as the text.
Your task is to generate one
challenging question and one
```

corresponding answer that includes
additional reference element
information, following these rules:
-----------------------------------
**Rule1:** Question must combine
multiple semantic elements from the
given data.
**Rule2:** Include image-based clues
indirectly (e.g., "the person in the
photo") instead of explicit names
from the text, and align them with
the text content.
**Rule3:** No invented or contradictory
facts.
**Rule4:** Be concise: single question
only, brief answer (word/phrase), and
answer must come from semantic info,
not directly from the image.
**Rule5:** The reference dict's key =
element fragment in the question,
value = its "name" from the JSON.

### Prompt for Query Generation

Now, based on the given image
and following text, generate one
challenging question and one
corresponding answer in English that
satisfy the rules above, and output
them in structured JSON form.
**Title:**
{title}
**Elements**
{element}

### System Prompt for Multi-Hop Evidence Generation

You are a evidence generation
assistant. You will be given:
– An image.
– A list of extracted semantic
elements.
– A set of question-answer pairs.
– A Wikipedia-style textual
description.
Your task is to generate five
evidence sentences, each
approximately 50 words long,
following the rules below:
-----------------------------------
**Rule1:** Output exactly 5 evidence
sentences/short paragraphs ( 50 words
each).
**Rule2:** At least 1 item must be
factually consistent with the image,
semantic elements, and the given
answer (supporting evidence)
**Rule3:** Other items should replace
some semantic elements from the list
to introduce subtle conflicts while

staying generally relevant to the
image and question. Keep context
coherent.
**Rule4:** You may add minor background
info from the Wikipedia text to reach
length, but the core must relate to
the image and question.
**Rule5:** Evidence should imitate
Wikipedia's descriptive and
explanatory tone, without words like
"comparison," "summary," or "as shown
in the image."
**Rule6:** Do not mention differences
between original and rewritten
content.

### System Prompt for Common Attribute Evidence Generation

You are a question refinement and
evidence generation assistant. You
will be given:
– An image.
– A list of extracted semantic
elements.
– A set of question{answer pairs.
– A Wikipedia-style textual
description.
Your task has three steps:
1. Rewrite the given question into a
shared-attribute question.
2. Generate the corresponding answer
to this rewritten question.
3. Generate five evidence sentences
($\approx$50 words each) for the rewritten
question and answer.
following the rules
below, especially **rule8:**
-----------------------------------
**Rule1:** A shared attribute means a
general element or property from
the semantic list (avoid specific
names/values).
**Rule2:** Do not use phrases like
"shared feature" explicitly-let it
be inferred.
**Rule3:** Exactly 5 evidence items.
**Rule4:** Each evidence =
natural-language sentence/short
paragraph ( 50 words).
**Rule5:** $\geq$ 1 evidence must match the
image, semantic elements, and the
answer (**supporting evidence**)
**Rule6:** Others should subtly conflict
by replacing entries from the
semantic list, but remain relevant
and coherent.
**Rule7:** Small amount of background
info from the Wikipedia text allowed
to reach length; core must still
relate to topic.
**Rule8:** Evidence should imitate

```
Wikipedia's descriptive/explanatory
style|no "comparison," "summary," or
"shown in the image."
Rule9:  Do not mention differences
between rewritten and original.
```

### Prompt for Evidence Generation

```
Now, based on the given image and
following textual information,
generate five evidence sentences that
satisfy the rules above, and output
them in structured JSON form.
Please remember that the generated
evidence must include the key
semantic information of the
replaced element that is crucial
for supporting the answer, and
the evidence must not contain any
comparison between the information
before and after the replacement.
Elements:
{element}
Question:
{question}
Answer:
{answer}
Description:
{description}
```

### System Prompt for Data Quality Evaluation

```
You are an expert in evaluating
the annotation quality of
knowledge-conflict KI-VQA data.  For
each sample, you will be given:
- An image.
- A set of related QA pairs.
- A list of key semantic elements.
- A set of 5 constructed evidence
passages.
Your task is to assess the overall
data quality of this sample and
output a single quality score
in the range 0--10(integer).
------------------------------------
Rule1:  Image must be relevant to the
QA pairs.
Rule2:  Question must avoid directly
naming specific objects/text from
the image.  Use higher-level category
references (e.g., "the animal in the
image").
Rule3:  The 5 evidence passages must
have meaningful conflicts related
to key semantic elements.  Evidence
for the same element must differ
semantically (attributes, categories,
relations) while remaining plausible.
Rule4:  Evidence should read in
fluent, natural, Wikipedia-style
```

```
prose|no template-like replacements
or awkward text.
------------------------------------
Scoring Guidelines:
0-3:  Data quality is very poor,
with major rule violations such as
irrelevance, no real conflict, or
non-Wikipedia style.
4-6:  Data quality is medium, with
partial rule compliance and clear
issues in relevance, conflict, or
style.
7-8:  Data quality is good, with most
rules met and only minor flaws.
9-10:  Data quality is excellent,
with all rules met, strong conflicts,
and fluent Wikipedia-style evidence.
```

### D.2. Prompts for Discriminator

### Prompt for Direct Inference

```
You are an expert in detecting
conflicts among external knowledge
passages for visual question
answering.
For each sample, you will be given:
- An image.
- One natural-language question about
the image.
- Several external knowledge passages
that are intended to help answer the
question.
Your task is to determine whether
there are key information conflicts
among these external knowledge
passages with respect to the question
and the information implied by the
image.
------------------------------------
- Output `"yes"` if there is at least
one meaningful conflict between the
external knowledge passages under the
above definition.
- Output `"no"` if there is no
meaningful conflict and the passages
are mutually compatible (or only
differ in minor, non-essential ways).
```

### Prompt for Few Shot Inference

```
You are an expert in detecting
conflicts among external knowledge
passages for visual question
answering.
For each sample, you will be given:
- An image.
- One natural-language question about
the image.
- Several external knowledge passages
```

```
that are intended to help answer the
question.
Your task is to determine whether
there are key information conflicts
among these external knowledge
passages with respect to the question
and the information implied by the
image.
You must strictly follow the rules
below.
─────────────────────────────────
Rule1:  Judge conflicts only on
key semantic information relevant
to answering the question (e.g.,
entities, attributes, relations,
numbers, locations, categories).
Rule2:  If ≥ 2 passages make mutually
incompatible statements about such
key information, treat it as a
conflict.
Rule3:  Ignore minor wording/detail
differences or extra non-essential
background that do not change the
core fact.
Rule4:  Use the image and question
only for context in deciding what
is key; judge conflicts among the
passages themselves, not against the
image.
Rule5:  Final output:  one word only.
– Output '"yes"' if there is at least
one meaningful conflict between the
external knowledge passages under the
above definition.
– Output '"no"' if there is no
meaningful conflict and the passages
are mutually compatible (or only
differ in minor, non-essential ways).
```

**Prompt for RPA-SFT**

```
You are an expert in detecting
conflicts among external knowledge
passages for visual question
answering.
For each sample, you will be given:
– An image.
– One natural-language question about
the image.
– Several external knowledge passages
that are intended to help answer the
question.
Your tasks are:
1.  Decide whether there are
meaningful conflicts among these
external knowledge passages.
2.  Identify the key semantic
information units that appear in
the question and in the external
knowledge passages, and that are
directly involved in the reasoning
process needed to answer the
```

```
question.
```

## D.3. Prompts for Decoding

**Prompt for RPGD Decoding**

```
You are an expert visual question
answering model that jointly uses the
image and textual evidence to answer
questions.  Answer the question using
a single word or phrase.
For each sample, you will be given:
– An image.
– One natural-language question about
the image.
– Several external knowledge passages
that are intended to help answer the
question.
Your task is to answer the question by
combining information from the image
and the evidence passages, while
carefully handling conflicts.
```

**Prompt for Greedy Decoding (E-VQA)**

```
Context:  {passages}
Question:  {question}
Answer the question using a single
word or phrase.
The answer is:
```

**Prompt for Greedy Decoding (InfoSeek)**

```
You always answer the question the
user asks.  Do not answer anything
else.
──────────────────────────
Example
Context:  The sounthern side of the
Alps is next to Lake Como.
Question:  Which body of water is
this mountain located in or next to?
Short answer is:  Lake Como
──────────────────────────
Context:  {passages}
Question:  {question}
Short answer is:
```

## E. Comprehensive Case Studies

In this section, we present detailed case studies to illustrate the performance of our approach. The current RPGD decoding method already improves answer accuracy significantly, yet certain limitations remain, as highlighted in the failure cases. These observations provide valuable insights for future method refinement.

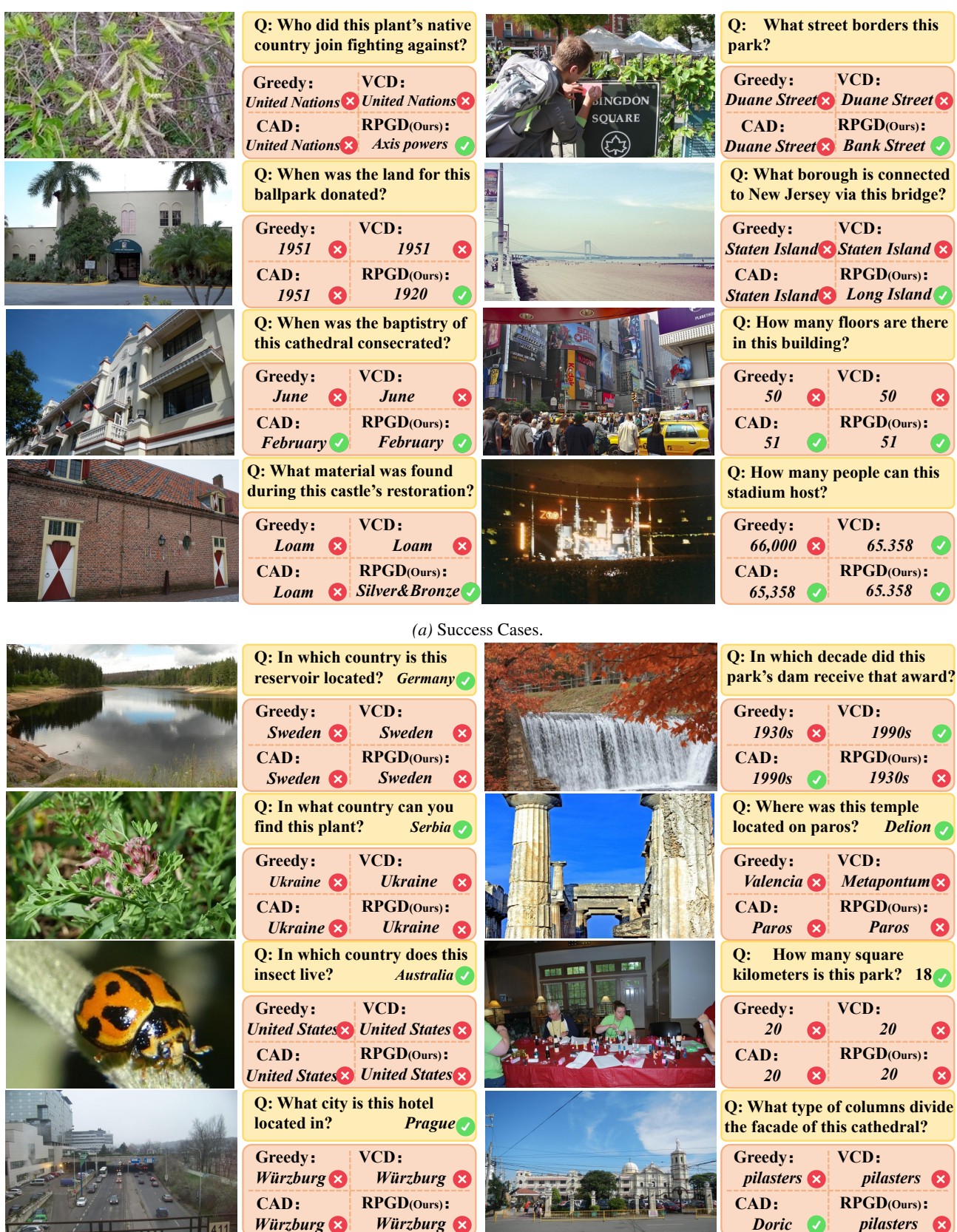

*Figure 8.* Representative success and failure cases from the **E-VQA** dataset, illustrating decoding methods across different scenarios.

