# OpenReview forum: "REAL: Resolving Knowledge Conflicts in Knowledge-Intensive Visual Question Answering via Reasoning-Pivot Alignment"
_ICML.cc/2026/Conference — ICML 2026 regular_

### Official Review · Reviewer_kY9s · 2026-03-04

**Soundness:** 3
**Presentation:** 4
**Significance:** 3
**Originality:** 3
**Overall Recommendation:** 3
**Confidence:** 3

**Summary:**

This paper focuses on a practical but hard issue in knowledge-intensive VQA (KI-VQA): open-domain retrieval often returns evidence that conflicts with parametric knowledge or even with the image, and this conflict can reliably derail multi-hop reasoning. The paper proposes REAL, which centers a “reasoning pivot” and defines a conflict as mutually exclusive assertions within the same pivot’s information scope, aiming to avoid overly broad conflict signals from entity/keyword mismatch. The method has two parts. First, RPA-SFT uses explicit `<RPivot>` span tagging and a multi-stage supervision pipeline (question pivot extraction → paragraph pivot localization → conflict verification) so the model learns pivot-aware conflict discrimination. Second, RPGD is a training-free contrastive decoding method that builds a “conflict-dominant pathway” via patch shuffling and then reduces conflict interference using adaptive gating and projection-based suppression. The paper introduces the REAL-VQA dataset and reports improvements on InfoSeek, E-VQA, and A-OKVQA with ablations and decoding comparisons.

**Compliance With Llm Reviewing Policy:**

Affirmed.

**Key Questions For Authors:**

1. RPGD constructs the “conflict-dominant pathway” via patch shuffling. Could you clarify whether you evaluate potential side effects of this design—since shuffling disrupts image information, it may weaken the role of visual evidence during reasoning and introduce effects unrelated to conflicts (e.g., more general visual degradation or changes in vision–language fusion)? What controls or diagnostics do you use to support that this pathway mainly reflects conflict-specific interference?

2. On attribution of end-to-end KI-VQA gains: the paper notes that performance depends on retrieval quality and pivot precision. Could you clarify whether you quantify the contribution of different error sources (e.g., retrieval failure vs. pivot extraction failure vs. conflict mitigation failure) and provide a more systematic case analysis? This breakdown helps support the causal link between conflict handling and the reported accuracy gains.

3. On the pivot definition vs. supervision: the paper defines a pivot as an “indispensable minimal set of nodes/edges under a zero-prior assumption,” but the data annotation and training pipeline relies on LLM-produced span-level `<RPivot>` tags. Could you clarify what in your annotation pipeline justifies that these LLM tags actually reflect *indispensability* and *minimality*, and to what extent they support the idea that the resulting pivot set is unique?

4. Also, do different LLMs (or different sampling settings of the same LLM) produce pivot sets that are semantically equivalent but structurally different? If so, how does this annotation variance affect training stability, evaluation consistency, and reproducibility?

**Limitations:**

yes

**Strengths And Weaknesses:**

**Strengths**

1. The paper targets a real failure mode in retrieval-augmented MLLMs: conflicting retrieval evidence is not rare, and it systematically hurts multi-hop reasoning; the motivation is convincing and practically relevant.

2. The pivot-centric conflict definition narrows conflicts to mutually exclusive statements within the same pivot scope, which plausibly reduces noisy “conflict” signals from superficial entity/keyword mismatch and better matches the structure of KI-VQA reasoning.

3. RPA-SFT places supervision on intermediate reasoning structure through explicit `<RPivot>` tags and multi-stage tasks, instead of relying on a single conflict label, which makes the training signal more aligned with “where/why” the conflict happens.

4. RPGD provides a relatively structured inference-time intervention: it does not simply subtract a reference logit vector, but constructs a second pathway, decomposes conflict-aligned components via projection, and applies adaptive gating on pivot-related tokens, which leaves room for interpretability and diagnosis.

5. The experiments cover both end-to-end KI-VQA accuracy and conflict discrimination metrics (F1/MCC), and they include comparisons against VCD/CAD-style decoding and component ablations, which helps readers understand whether gains come from training or decoding.

**Weaknesses**

1. The paper does not analyze whether RPGD gating harms normal tokens when pivot identification is wrong; this failure mode is plausible in realistic retrieval and multi-step reasoning settings, but the paper does not provide systematic robustness evidence (e.g., controlled pivot perturbations, gate coverage/false-suppression rates, or failure-case attribution) to support stability under noisy pivots.

2. RPGD depends on multiple hyperparameters (ε, β, κ, τ, δ) and on the specific mapping from pivot spans to token index sets, but the paper does not present sensitivity analysis or principled default choices, which makes it less clear how reliably the method transfers across backbones, retrieval configurations, and decoding setups.

3. RPGD constructs the “conflict-dominant pathway” via patch shuffling, but this inference construction disrupts image information and can weaken the role of visual evidence during reasoning, potentially introducing additional effects beyond conflicts (e.g., general visual degradation or changes in vision–language fusion). As a result, the second-path logits can mix conflict-related interference with effects induced by corrupted visual input, and the current evidence does not show that it consistently isolates conflict-specific interference.

4. The experiments mainly cover models up to ~8B parameters under specific retrieval settings, so it remains unclear how well the approach generalizes to larger or stronger MLLMs. Larger models often have richer parametric knowledge and different tendencies in how they weigh retrieved evidence, so more direct evidence is needed to clarify the external validity and practical scope of the method.

---

> ### Author Rebuttal · Authors · 2026-03-31
>
> We sincerely appreciate your constructive and insightful comments. We will explain your concerns point by point.
>
> **Q1: Stability under Noisy Pivots.**
>
> **A1**: Thank you for your question. Our adaptive gating and orthogonal design avoid over-suppression. Tokens with naturally low conflict scores are barely inhibited. We test robustness by randomizing 50% stage-1 pivot outputs and measuring the False Suppression Rate (FSR). Experimental results are shown below:
>
> | Method | Guidance Signal| FSR | E-VQA SH| E-VQA All|
> |---| ---| ---| ---| ---|
> | Qwen3-VL-8B |/| /| 42.4| 38.1|
> | REAL (Ours)| Random Signal| 1.87| 43.5|39.3|
> | **REAL (Ours)**| **Stage1 Signal**|**0.32**|**45.4**|**41.4**|
>
> **Q2: Hyperparameter Settings.**
>
> **A2**: We are sorry for the unclear details. We use a fixed optimal set (**ε = 0.1, β = 0.2, κ = 0.1, δ = 10⁻⁶**) in Appendix B.3 across all models.Tested ranges (ε ∈ [0.05, 0.2], β ∈ [0.1, 0.3], κ ∈ [0.05, 0.2]) also improve baselines with smaller gains, balancing suppression and stability.
>
> **Q3: Validation on Lager Models.**
>
> **A3**:Table 10 (Line 825) shows our conflict discriminator brings a **12.1%** average gain on Qwen3-VL-32B. The full REAL pipeline also outperforms all baselines, demonstrating its strong generalization to larger MLLMs. Experimental results are shown below:
>
> |Method|E-VQA SH|E-VQA All|
> |---|---|---|
> | Qwen3-VL-32B|24.6|24.7|
> | + Knowledge|48.6|44.9|
> |**+ REAL (Ours)**|**51.5**|**47.6**|
>
> **Q4: Side Effects of Patch Shuffling.**
>
> **A4**: We validate the design with two experiments. First, under oracle passage supervision, patch shuffle causes almost no performance loss, so it **does not disturb vision–language fusion**.
>
> | Model | Input| E-VQA SH | E-VQA All| IS UQ| IS UE| IS All|
> | --- | --- | --- | --- | --- | --- | ---|
> | Qwen3-VL-2B | image| 16.0| 15.6| 14.6| 13.3| 14.0|
> | Qwen3-VL-2B| image + oracle text| 54.3| 49.7| 57.2| 56.7| 56.9|
> | REAL-Qwen3-VL-2B(Ours)| image + oracle text| 54.2| 49.4| 57.2| 56.6| 56.9|
>
> Second, unlike blanking and Gaussian blur that destroy global visual information, the model tends to rely more on internal parametric knowledge instead of trustworthy external text when visual cues become unreliable[1]. Our method only disrupts local patterns, allowing the model to still trust external knowledge while focusing more on textual conflict signals than visual cues.
>
> | Method | Visual| E-VQA SH| E-VQA All | IS UQ | IS UE | IS All |
> | --- | --- | --- | --- | --- | --- | --- |
> | Qwen3-VL-8B | /| 42.4 | 38.1 | 38.0 | 39.4 | 38.7 |
> | + RPGD| Blank| 40.8 | 36.9 | 36.8 | 37.9 | 37.4 |
> | + RPGD| Gaussian| 42.2 | 38.1 | 38.6 | 41.2 | 39.9 |
> | **+ RPGD** | **Patch Shuffle** | **45.4**| **41.4**| **43.1**| **45.1**| **44.1**|
>
> **Q5: Error Causal Analysis.**
>
> **A5**: We analyze error sources across retrieval, pivot extraction (Stage 1), conflict mitigation (Stage 2), and VL reasoning. The table gives the **percentage breakdown of each error type**. For retrieval-correct samples, the two stages each resolve ~40% of remaining errors. Detailed distribution of errors is as follows:
>
> | Dataset| Retrieval| Pivot Extraction| Conflict Mitigation| VL Reasoning|
> | :-: | :-: | :-: | :-: | :-: |
> |  E-VQA |46%|22%|16%|16%|
> | InfoSeek |21%|25%|19%|35%|
>
> This distribution reveals that besides retrieval, the model’s own reasoning capacity is also limited, and RPGD resolves most issues when provided with accurate pivot extraction.
>
> **Q6: Pivot Definition vs. Supervision.**
>
> **A6**: We ensure LLM-generated `<RPivot>` meets key properties via a structured pipeline:**1)Indispensability**:LLM + expert review keeps only reasoning-critical spans.**2)Minimality**:Following a question-first strategy, we anchor passage annotations to the pre-labeled question pivots, which effectively reduces redundancy.3)These two constraints jointly limit valid reasoning paths, leading to **unique** and highly consistent pivot sets
>
> We compare against paragraph-first labeling, and the results confirm that our approach achieves **higher accuracy and lower redundancy**, validating the effectiveness of our annotation design.
>
> | Annotation| Avg. Extracted| Avg. Gold| F1|
> | ---| ---| ---| ---|
> | Paragraph-first| 4.8| 3.0| 76.6%|
> | **Question-first(Ours)** | **3.2**| **2.9**| **93.3%**|
>
> **Q7: LLM Annotation Variance.**
>
> **A7**: We agree that structurally different pivots exist, but their impact is limited. Core conflict terms remain **highly consistent**, as annotation uses the question as reference. We further train Qwen3-VL-8B on annotations from another LLM, evaluate inter-annotation consistency and validate stability on both ground-truth sets.
>
> | Model | Source| Avg. Pivot| Overlap| F1 (GT: 4o) | F1 (GT: Qwen3-VL) |
> | ---| ---| ---| ---| ---| ---|
> | Qwen3-VL-8B | GPT-4o | 3.2| 81.8% |72.0| 70.7|
> | Qwen3-VL-8B | Qwen3-VL-235B-A22B| 3.0| 90.1%| 70.3| 71.3|
>
> The token-level suppression in Stage 2 RPGD also ensures stable effectiveness.
>
> [1] Xu, R., et al. Knowledge Conflicts for LLMs: A Survey. 2024.

---

> > ### Author Rebuttal · Reviewer_kY9s · 2026-04-06
> >
> > Thanks for the clarification. I hope this discussion helps make your paper even stronger.

---

> > > ### Author Response · Authors · 2026-04-06
> > >
> > > We sincerely appreciate your support and valuable comments. These questions have greatly helped us strengthen the paper from several key perspectives, including the design of the conflict-dominant pathway, the attribution of performance gains, and the reliability of pivot annotation and supervision. We will incorporate these improvements into the next version of the paper. Should our responses resolve all your concerns, we would be deeply grateful for your kind consideration in re-evaluating our work.

---

### Official Review · Reviewer_R2AV · 2026-03-08

**Soundness:** 3
**Presentation:** 3
**Significance:** 2
**Originality:** 2
**Overall Recommendation:** 3
**Confidence:** 4

**Summary:**

The paper proposes REAL, a framework designed to handle knowledge conflicts in Knowledge-Intensive Visual Question Answering (KI-VQA). The authors introduce the "Reasoning-Pivot" concept to represent atomic entities and relations within a reasoning chain. The main contributions include a synthetic dataset (REAL-VQA), a supervised fine-tuning strategy (RPA-SFT) that leverages special tokens to extract pivots, and a contrastive decoding method (RPGD). RPGD attempts to isolate conflict interference by contrasting standard logits with those generated from an image patch-shuffling pathway, using adaptive gating and Gram-Schmidt orthogonalization to adjust the final output distribution.

**Compliance With Llm Reviewing Policy:**

Affirmed.

**Final Justification:**

I have carefully reviewed the authors' rebuttal along with the other reviewers' comments. The rebuttal addressed some of my concerns but did not fully resolve the core issues I raised. After weighing the strengths and weaknesses in terms of soundness, originality, significance, and clarity, I maintain my original assessment and keep my score unchanged.

**Key Questions For Authors:**

1. What is the fundamental theoretical distinction between your "Reasoning-Pivot" and standard entity/relation extraction in multi-hop QA computational graphs? Please explicitly compare your theoretical contribution against recent 2024-2025 literature on atomic knowledge decomposition in RAG.

2. How does degrading visual topology via patch shuffling logically assist the model in resolving a pure text-to-text knowledge conflict within the retrieved documents? If the visual context serves as the factual anchor, why wouldn't destroying it exacerbate hallucination?

3. GPT-4o generation pipeline. Can you provide evaluation metrics on human-annotated, naturally occurring knowledge conflicts to prove the model has not simply overfit to specific GPT-4o text generation artifacts?

**Limitations:**

yes

**Strengths And Weaknesses:**

**Strengths:**

Soundness of Decoding: The application of Gram-Schmidt orthogonalization in RPGD is mathematically rigorous. It presents a more targeted approach to isolating specific feature vectors during contrastive decoding compared to the blunt linear scalar subtraction utilized by baselines like VCD and CAD.

Presentation: The problem of retrieval-induced knowledge conflicts in multimodal systems is highly relevant, and the pipeline is presented clearly.

**Weaknesses:**

Rebranding and Limited Novelty: The core "Reasoning-Pivot" concept is a blatant rebranding of standard multi-hop knowledge graph reasoning, specifically entity and relation extraction. Decomposing context into atomic facts or sub-claims to resolve conflicts in RAG systems is an established standard in the literature (e.g., HopRAG for logic-aware multi-hop reasoning, or the CLEAR framework for probing latent knowledge conflicts at the sentence level). Appending <RPivot> tags to entities during training is a basic prompt engineering and fine-tuning tactic. It lacks fundamental architectural innovation.

Motivation-Method Disconnect: The RPGD strategy relies on visual patch shuffling to construct a "conflict-dominant" pathway. However, knowledge conflicts in KI-VQA are primarily driven by contradictory external text retrieved by the RAG system. Degrading the visual structure does not help the model resolve textual contradictions; it simply forces the model to rely entirely on the potentially flawed text. The reasoning behind this specific perturbation is disconnected from the nature of the text-based conflict it aims to solve.

Circular Evaluation Setup: To evaluate conflict discrimination on E-VQA and ScienceQA, the authors synthesize conflict samples using the exact same GPT-4o pipeline used to create their training set. Testing the RPA-SFT model on synthetic distributions that perfectly match its training data artificially inflates the performance metrics in Table 3 and fails to prove true out-of-distribution generalization.

---

> ### Author Rebuttal · Authors · 2026-03-31
>
> We sincerely appreciate your professional comments. We address each of your concerns in detail below.
>
> **Q1: Rebranding and Limited Novelty.**
>
> **A1**: We appreciate your critical comments. Our Reasoning-Pivot targets fine-grained external conflict resolution, with different application scenarios and design goals from typical RAG approaches.
>
> - **Granularity**: We operate at token level to capture local word-level conflict knowledge, whereas HopRAG[2] and CLEAR[1] rely on coarse passage or sentence units that are widely adopted in conventional RAG systems[4].
> - **Scope**: Some existing methods do use NER to extract entities for graph construction, yet they extract all entities and introduce substantial irrelevant noise into constructed graphs[4]. By contrast, we follow the minimal necessary principle and retain only tokens critical to reasoning. Retrieved passages in KI-VQA contain extensive static information irrelevant to the question.
> - **Conflict type**: Prior works[1,3] target conflicts between external text and parametric knowledge. Accordingly, these works require knowledge to be split into sentence forms consistent with parametric representations. Our method instead resolves contradictions between external knowledge sources expressed as key token-level conflicts.
> - **Stage**: Existing approaches mainly focus on improving retrieval efficiency[2] or regularizing model training[1], which operate at the retrieval or prompt-engineering stage. By comparison, our pivot serves as a dynamic inference trigger and directly intervenes in the generation decoding phase.
>
> As stated in Section 4.2, although tagging is a common technique, we introduce the `<RPivot>` token to better guide the model in mimicking human reasoning steps for conflict resolution: extract pivots from questions, locate them in context, and perform verification on these pivots.
>
> **Q2: Motivation-Method Disconnect.**
>
> **A2**: Thank you for this rigorous discussion. Knowledge conflicts appear in text but are not caused by text alone. Visual cues are vital for multimodal reasoning. In KI-VQA, reasoning is visually grounded, with context, entities and chains anchored to the image. We test visual guidance by comparing full-image and caption inputs with fixed questions and knowledge. The table below validates that without visual information cues, the model leans more on parametric knowledge[4], yielding unstable logit ratios between correct and conflicting answers, where the edge of correct tokens fades.
>
> | Input  | Correct / Conflict Logit Ratio | Std  |
> | --- | --- | --- |
> | Image + Knowledge + Question | 1.62 | 0.31 |
> | Image Caption + Knowledge + Question | 1.07 | 0.86 |
>
> Our patch shuffle breaks fine spatial structures while preserving global image semantics, so the model still retains the overall information of the image, with only a slight weakening of visual verification.By contrast, blank masking removes visual input entirely and forces the model to be driven by internal parameters, while Gaussian blur only retains limited global information, placing its effect between our method and blank masking. Results below verify the effectiveness of our approach.
>
> | Method      | Visual Processing | E-VQA SH | E-VQA All | InfoSeek UQ | InfoSeek  UE | InfoSeek  All |
> | --- | --- | --- | --- | --- | --- | --- |
> | Qwen3-VL-8B | / | 42.4 | 38.1  | 38.0 | 39.4| 38.7|
> | + RPGD| Blank | 40.8 | 36.9 | 36.8 | 37.9  | 37.4|
> | + RPGD | Gaussian | 42.2 | 38.1 | 38.6| 41.2 | 39.9 |
> | **+ RPGD**  | **Patch Shuffle** | **45.4** | **41.4**  | **43.1** | **45.1** | **44.1** |
>
> **Q3: Circular Evaluation Setup.**
>
> **A3**: Thank you for your question. Our method has been validated on MMKC, a standard benchmark with high-quality expert annotations, achieving **25.3%** and **8.9%** gains over Qwen3-VL and LLaVA respectively as shown in Table 3. For better generalization, we also test on two real-world conflict datasets: e-SNLI-VE with native contradiction labels and infoseek-human with manual annotations.
>
> | Model        | e-SNLI-VE | InfoSeek-human |
> | --- | --- | --- |
> | Qwen3-VL-8B  | 71.6 | 63.4 |
> | + RPA-SFT    | **80.5**  | **75.0** |
> | LLaVA-1.5-7B | 62.1| 50.3  |
> | + RPA-SFT    | **73.6**  | **66.9** |
>
> We also analyze conflict types in E-VQA and ScienceQA. Even with a similar pipeline, their conflict distributions are highly distinct.
>
> | Dataset   | Source    | Factual | Attribute | Logical | Arithmetic | Conceptual |
> | ---| ---| ---| ---| ---| ---| ---|
> | ScienceQA | Textbooks | **✓**| ✗| **✓**| **✓** | **✓** |
> | E-VQA | Wikipedia | **✓**   | **✓** | ✗ | ✗ | ✗|
>
> [1] Gao, L., et al. Probing Latent Knowledge Conflict for Faithful Retrieval-Augmented Generation. 2025.
>
> [2] Liu, H., et al. HopRAG: Multi-Hop Reasoning for Logic-Aware Retrieval-Augmented Generation. 2025.
>
> [3] Zhang, Q., et al. FaithfulRAG: Fact-Level Conflict Modeling for Context-Faithful Retrieval-Augmented Generation. 2025.
>
> [4] Xu, R., et al. Knowledge Conflicts for LLMs: A Survey. 2024.

---

> > ### Author Rebuttal · Reviewer_R2AV · 2026-04-01
> >
> > We thank the authors for their detailed rebuttal.
> >
> > **Q1 (Novelty):** The four dimensions listed (granularity, scope, conflict type, stage) describe engineering-level differences rather than a fundamental theoretical distinction. Wrapping entity/relation spans with `<RPivot>` tokens is still span-level extraction, and the "minimal necessary principle" closely mirrors standard question decomposition in multi-hop QA. The core concern remains unaddressed.
> >
> > **Q2 (Patch Shuffle Motivation):** The new ablation comparing Blank/Gaussian/Patch Shuffle is appreciated and demonstrates empirical superiority. However, the logical gap persists: if the image serves as the factual anchor for resolving textual contradictions, degrading it should weaken—not strengthen—the model's ability to adjudicate text-vs-text conflicts. The rebuttal shows that patch shuffle *works better than alternatives*, but does not explain *why* destroying visual topology should help resolve contradictions that originate entirely in retrieved text. This remains an "it works" argument, not a "why it should work" one.
> >
> > **Q3 (Circular Evaluation):** The additional results on MMKC (expert-annotated), e-SNLI-VE (native contradiction labels), and InfoSeek-human (manual annotations) meaningfully address this concern. We acknowledge these as valid evidence of out-of-distribution generalization.
> >
> > Given the partial resolution of Q3, we raise our overall score from **2 → 3**. However, the unresolved issues in Q1 (limited conceptual novelty) and Q2 (motivation-method disconnect) continue to weigh on our assessment.

---

> > > ### Author Response · Authors · 2026-04-02
> > >
> > > We sincerely appreciate your timely feedback and constructive comments, which help us further clarify the core contributions of our work and address the remaining concerns.
> > >
> > > **Q1: Novelty.**
> > >
> > > **1.Span Extraction.** As discussed in our *Scope* section, although our extraction shares similar granularity with some existing methods, **selectively extracting reasoning-related entities reflects a theoretical difference rather than merely an engineering optimization**. Methods typified by GraphRAG[1] perform full extraction of all possible entities and relations to build comprehensive graphs, leading to **high computation and information redundancy**. By contrast, our approach only extracts entities directly involved in the reasoning chain, which fundamentally resolves these key drawbacks.
> > >
> > > **2.Multi-hop Question Decomposition.** In knowledge conflict scenarios, standard question decomposition and our pivot-based partitioning follow distinct logic. Standard multi-hop methods decompose a question into a sequence of dependent steps based on logical connections. Once conflicting text appears in an early step, **errors propagate along the chain and break the entire logical path**. As highlighted in Section 4.2, our method identifies Reasoning Pivots by **matching the intrinsic reasoning structure of the original question**, rather than relying on hierarchical and progressive decomposition. This design effectively avoids cascading failures and maintains stable reasoning under conflicting knowledge, supporting cases that standard sequential decomposition cannot handle.
> > >
> > > **Q2: Patch Shuffle Motivation.**
> > >
> > > Thank you for acknowledging the empirical superiority of our patch shuffle method as demonstrated in the new ablation study.
> > >
> > > Prior studies have verified that KI-VQA requires joint image-text understanding to construct valid reasoning chains. To explain *why* Patch Shuffle works, we summarize the model’s behavior in three stages according to visual credibility:
> > >
> > > - Stage1: When the **image is intact and highly reliable**, the model uses vision as the initial anchor for reasoning and incorporates textual information as support. In such cases, knowledge conflicts exist as one source of errors, yet **their contribution to reasoning failures is difficult to isolate and quantify**.
> > > - Stage2: After moderate Patch Shuffle, global visual semantics remain identifiable but fine-grained spatial details become unreliable. The model still regards external information as a valid reference, but relies **more heavily on textual content**. Its ability to **judge which option in conflicting text is more consistent with visual information decreases** as vision-text alignment capability weakens. Meanwhile, **the relative influence of conflicting text choices on model reasoning increases**, and this portion of interference is **exactly what we aim to remove**.
> > > - Stage3: This does not mean visual influence can be **reduced indefinitely**. When visual credibility is extremely low, such as under Blank or Gaussian corruption, the model can no longer treat vision as a trustworthy reference. Since the retrieved text already contains contradictions, the model tends to rely primarily on internal parametric knowledge, manifested as outputs unrelated to external multimodal cues.
> > >
> > > Our standard reasoning path corresponds to the first stage, and the conflict-dominant path operates in the second stage. Using gating and orthogonal projection, we **eliminate the negative impact of knowledge conflicts extracted from the conflict-dominant path on the standard reasoning path**, enabling cases originally failed due to such conflicts to be answered correctly.
> > >
> > > To verify the model’s information reliance across the three stages, we generate conflicting entity candidates using the `wikipedia_url` field from E-VQA retrieval results and perform progressive Patch Shuffle experiments with Qwen3-VL-8B to analyze the effects of visual degradation.
> > >
> > >
> > > |Patch Setting| Input| Correct Entity (%)| Conflicting Entity (%)| Unrelated Entity (%)|
> > > | -| -| -| -| -|
> > > |1×1|Image-only|95.0|/|/|
> > > |1×1|Image + Conflicting Text|83.5|13.2| 3.3|
> > > |2×2|Image-only| 94.3|/|/|
> > > |2×2|Image + Conflicting Text| 81.2| 15.1| 3.7|
> > > |3×3|Image-only|83.4|/|/|
> > > |3×3|Image + Conflicting Text| 64.1|30.6|5.3|
> > > |4×4|Image-only|65.2|/| /|
> > > |4×4|Image + Conflicting Text| 42.7| 26.9|30.4|
> > >
> > > Within the 1×1 to 3×3 settings, global visual semantics maintain high reliability, and the model can **accurately identify visual content**. However, as shuffle granularity increases, the **proportion of answers biased toward conflicting text gradually rises**. When the setting exceeds 4×4, visual credibility degrades significantly, leading to **a sharp increase in unrelated answers** dominated by the model’s internal parametric knowledge.
> > >
> > > Finally, we thank you for your suggestions and will incorporate them into the next version.
> > >
> > > [1] Edge, D., et al. From Local to Global: A Graph RAG Approach to Query-Focused Summarization. 2024.

---

### Official Review · Reviewer_FV3a · 2026-03-10

**Soundness:** 3
**Presentation:** 3
**Significance:** 3
**Originality:** 3
**Overall Recommendation:** 4
**Confidence:** 3

**Summary:**

This paper studies the problem of knowledge conflicts in Knowledge-Intensive Visual Question Answering (KI-VQA). When a model retrieves external passages to answer questions, the retrieved documents may contain mutually contradictory information, which can mislead the final answer. To address this, the authors propose a framework called REAL, built around the concept of "Reasoning-Pivot." The overall pipeline has two stages. The first stage trains a discriminator (RPA-SFT) that first extracts key reasoning nodes from the question and retrieved passages, and then performs binary conflict classification based on the consistency of those nodes. The second stage introduces a contrastive decoding strategy (RPGD) that constructs a "conflict-dominant path" via Patch Shuffle and contrasts it with the standard path using Gram-Schmidt orthogonal projection to suppress conflict-induced noise in the logits. The authors also construct a dataset, REAL-VQA (~4,700 samples), to support training and evaluation.

**Compliance With Llm Reviewing Policy:**

Affirmed.

**Key Questions For Authors:**

Please refer to the "Weaknesses" part.

**Limitations:**

Yes

**Strengths And Weaknesses:**

Strengths:

(1) The problem is well-motivated. Knowledge conflicts arising from imperfect open-domain retrieval are a real and often overlooked issue in KI-VQA.

(2) A new annotated dataset is introduced. REAL-VQA provides fine-grained annotations and structured external contexts, which could serve as a useful resource for future research on knowledge conflict in multimodal settings.

Weaknesses:

(1) The distinction between "Reasoning-Pivot" and standard entity/keyword extraction is not convincing enough.  The authors spend considerable effort arguing that their concept of reasoning-pivot is different from prior entity or keyword matching approaches. However, the first step of RPA-SFT is still a sequence tagging task that wraps key terms with special tokens.

(2) The core assumption of RPGD lacks justification. The logic of RPGD is as follows: Patch Shuffle is used to destroy the spatial structure of the image, which forces the model to over-rely on conflicting textual content, thereby constructing a "conflict-dominant path." The logits from this path are then orthogonally projected onto the standard path, and the aligned component is subtracted to remove conflict-induced interference. However, why should a path constructed by corrupting visual input reliably approximate the distributional shift caused by textual knowledge conflicts in the standard path? Patch Shuffle manipulates visual coherence, while the conflict noise in the standard path originates from semantic contradictions at the text level. There is no obvious reason why these two distributions should align well.

(3) The paper frames RPGD as being triggered by the conflict detection signal from Stage 1. However, since RPGD is a training-free strategy, it can be applied to all samples regardless of the detection outcome. It is unclear how much of the performance gain comes from the conflict detection signal versus the contrastive decoding mechanism itself. This decomposition is missing from the ablation analysis.

---

> ### Author Rebuttal · Authors · 2026-03-31
>
> Thank you for your insightful suggestions. We will explain your concerns individually in the following.
>
> **Q1: Reasoning-Pivot Definition.**
>
> **A1**: Thank you for your question. Although reasoning-pivot is similar in form to entity extraction, they differ fundamentally. As noted in [1], conventional extraction yields redundant spans that require extra post-processing and introduce noise. Prior methods lack a clear definition of critical reasoning roles and only extract general spans from the semantic surface, producing static units unrelated to inference. In contrast, reasoning-pivot focuses on logical inference chains and retains only core nodes essential to reasoning. We compare our method with information extracted by standard entity extraction, and the results show that **standard entity extraction yields only marginal improvements, whereas reasoning-pivot significantly boosts accuracy**, proving it is a reasoning-oriented structural unit rather than plain keywords.
>
> | Method | Pivot Type| Avg. Span | REAL-VQA | E-VQA | ScienceQA | MMKC |
> | -| - | -| -| -| - | - |
> | Qwen3-VL-2B | /  | / | 66.0| 64.5| 66.2| 64.8 |
> | REAL | Standard Entity | 7.8 | 71.2| 69.4  | 70.3| 64.3 |
> | **REAL(Ours)** | **Reasoning-pivot** | **3.2** | **94.2** | **82.8** | **80.0** | **69.3** |
>
> **Q2: RPGD Design Justification.**
>
> **A2**: Thank you for this rigorous discussion. Although knowledge conflicts appear in textual form, they are not caused by text alone. Visual information also plays a key role in the multimodal reasoning process. Without fine-grained visual cues, the model leans heavily on pre-trained parametric knowledge[1]. In KI-VQA, reasoning is visually grounded. The reasoning context, target entities and inference chains all start from the image. We verify the function of visual guidance by comparing two settings: full image input and image caption input, while keeping the question and external knowledge unchanged.
>
> Experimental results show that **when visual information is absent, the logit ratio between correct and conflict answers becomes unstable and hard to predict, and the clear advantage of correct tokens under visual guidance is no longer obvious**.
>
> | Input| Correct / Conflict Logit Ratio | Std  |
> | -| - | - |
> | Image + Knowledge + Question | 1.62 | 0.31 |
> | Image Caption + Knowledge + Question | 1.07| 0.86 |
>
> Patch shuffle preserves global image semantics while breaking fine spatial structures, so the model still maintains overall image information with only a mild weakening of visual verification. For ablation studies, blank masking fully removes visual input and forces the model to rely entirely on internal parametric knowledge. Gaussian blur retains only limited global visual information, yielding an effect intermediate between patch shuffle and complete visual removal. Experimental results below demonstrate that **our patch shuffle effectively targets and mitigates knowledge conflict bias in multimodal reasoning**.
>
> | Method      | Visual Processing | E-VQA SH | E-VQA All | InfoSeek UQ | InfoSeek  UE | InfoSeek  All |
> | - | - | - | - | - | - | - |
> | Qwen3-VL-8B | / | 42.4| 38.1| 38.0| 39.4| 38.7 |
> | + RPGD| Blank| 40.8 | 36.9| 36.8 | 37.9 | 37.4|
> | + RPGD| Gaussian | 42.2 | 38.1| 38.6 | 41.2 | 39.9|
> | **+ RPGD**  | **Patch Shuffle** | **45.4** | **41.4**  | **43.1**    | **45.1**| **44.1**|
>
> **Q3: Ablation on Conflict Detection Signal.**
>
> **A3**: As already verified in Table 5(line 361) and Table 7(line 385) of the paper, our RPGD decoding method yields consistent improvements across different models, with around 5% gain on InfoSeek and 3% gain on E-VQA. To isolate the performance contributions between conflict detection guidance and the contrastive decoding mechanism, we design three controlled variants based on our RPGD method: **1) REAL w/o Conflict Signal** applies contrastive decoding without any pivot guidance from conflict detection. **2) REAL w/ Random Signal** keeps the same number of pivots but randomly replaces 50% of them with irrelevant tokens. **3) REAL (Ours)** denotes the full REAL framework guided by accurate conflict detection signals from Stage 1. Results show **performance improves steadily as the guidance signal becomes more precise**. This confirms the gain comes from **both the contrastive decoding paradigm and reliable conflict detection**.
>
> | Method          | Guidance Signal      | E-VQA SH | E-VQA All | InfoSeek UQ | InfoSeek  UE | InfoSeek  All |
> | - | - | - | - | -| - | - |
> | Qwen3-VL-8B | /  | 42.4 | 38.1| 38.0 | 39.4 | 38.7 |
> | REAL-Qwen3-VL-8B | w/o Conflict Signal  | 42.7 | 38.1 | 38.4| 39.5| 39.0|
> | REAL-Qwen3-VL-8B | w/ Random Signal| 43.5 | 39.3| 39.6| 40.9| 40.3|
> | **REAL-Qwen3-VL-8B (Ours)** | **w/ Stage1 Signal** | **45.4** | **41.4**  | **43.1**| **45.1**  | **44.1**|
>
> [1] Xu, R., et al. Knowledge Conflicts for LLMs: A Survey. 2024.

---

> > ### Author Rebuttal · Reviewer_FV3a · 2026-04-03
> >
> > Most of my concerns have been addressed, and I currently have no other questions. I keep the rating.

---

> > > ### Author Response · Authors · 2026-04-03
> > >
> > > We sincerely appreciate your recognition and constructive comments. Your valuable feedback has greatly helped us improve our work, and we will incorporate these improvements in the next version.

---

### Official Review · Reviewer_ZFNr · 2026-03-12

**Soundness:** 3
**Presentation:** 3
**Significance:** 3
**Originality:** 3
**Overall Recommendation:** 4
**Confidence:** 4

**Summary:**

The paper proposes a reasoning-pivot centered knowledge-intensive VQA framework REAL, including the REAL-VQA dataset construction and sft, and a reasoning-pivot guided decoding mechanism. The paper aims to address the limited generalizability in knowledge conflict detection and absence of intra-model conflict handling challenges of existing KIVQA studies. Experiments are conducted on 3 datasets, demonstrating their effectiveness.

**Compliance With Llm Reviewing Policy:**

Affirmed.

**Final Justification:**

Most of my concerns have been addressed. I maintain my positive score.

**Key Questions For Authors:**

See Weaknesses.

**Limitations:**

Yes.

**Strengths And Weaknesses:**

Strengths:
1. The reasoning pivot mechanism is novel, and addresses an important challenge in KI-VQA.
2. The decoding mechanisms are well-designed. Experiments demonstrate the effectiveness of the components.
3. Experiments are comprehensive, covering KI-VQA accuracy and conflict discrimination. The evaluation also covers cross-domain conflict discrimination and demonstrates its generalization ability.

Weaknesses:
1. Parameter efficiency. Baselines such as VLM-PRF are based on LoRA tuning. REAL only freezes the visual encoders and trains the language backbone and projector layers. How does REAL perform using more parameter-efficient methods, such as widely-adopted LoRA?
2. Risk of error accumulation. What is the error accumulation situation in REAL? If conflict discrimination is wrong, would it affect the final reasoning? What is the RPGD performance if the oracle pivots or the discrimination signals are provided?
3. The RPGD mechanism essentially isolates and penalizes the text-dominant bias through visual disruption. This is based on an assumption that the text concluded without the whole image is the bias that needs to be eliminated. However, sometimes the image can be less meaningful, especially for KI-VQA where knowledge reasoning is more important, and the main reasoning is based on the text. In this scenario, it may harm the right reasoning procedure.
4. How do best-of-N/self-consistency/major-vote perform for decoding? Knowledge conflicts may be eliminated through multiple runs to stabilize to a consistent one.
5. Minor discussion: Detecting conflicts could be easy to accomplish, but the more interesting question here is which one is right or helpful knowledge for the reasoning. Discriminating and addressing the conflict seems a detour.

---

> ### Author Rebuttal · Authors · 2026-03-31
>
> We sincerely thank you for your insightful advice. We reply to your questions individually as follows.
>
> **Q1: Parameter Efficiency.**
>
> **A1**: We appreciate your question on parameter efficiency. To ensure fair comparisons with baselines like VLM-PRF, we applied LoRA to Qwen3-VL-8B using their **exact settings** (LoRA rank=64, LoRA alpha=128, dropout=0.05). This drastically reduced training time to just 5 hours. As shown below, this highly efficient LoRA setup **maintains comparable performance in both conflict detection F1 and final two-stage QA accuracy**.
>
> Table:Evaluation on Conflict Discrimination
>
> | Model | Tuning Method| REAL-VQA | E-VQA | ScienceQA | MMKC |
> | - | -| -| -| - | - |
> | Qwen3-VL-8B | Base (No Tuning) | 69.9 | 94.5 | 74.8| 66.9 |
> | Qwen3-VL-8B | Full Tuning| 99.1| 95.5  | 95.4| 74.8 |
> | Qwen3-VL-8B | LoRA| 97.3| 95.5  | 94.7| 74.2 |
>
> Table:Evaluation on VQA Accuracy
>
> | Method| Tuning Method | E-VQA SH | E-VQA All | InfoSeek UQ | InfoSeek UE | InfoSeek All |
> | -| -| -| -| -| -| -|
> | VLM-PRF| LoRA| 40.1| 39.2| 43.5| 42.1| 42.5 |
> | REAL(Ours) | Full Tuning| 45.4| 41.4| 43.1| 45.1| 44.1|
> | REAL(Ours) | LoRA | 45.2| 41.4| 43.0| 44.9| 44.0 |
>
> **Q2: Risk of Error Accumulation.**
>
> **A2**: Thank you for this question.We evaluate error accumulation by randomly replacing 50% of detected pivots to simulate real-world discrimination errors. We also report upper bounds using two oracle settings: only full conflict pivots and full reasoning pivots. Results show **our method performs close to oracle upper bounds. It degrades only slightly even under heavy pivot noise and still outperforms baselines, demonstrating negligible error accumulation**.
>
> | Method | Pivot Detection F1 | E-VQA SH | E-VQA All |
> | -| -| -| -|
> | Qwen3-VL-8B | / | 39.0 | 35.0 |
> | REAL-Qwen3-VL-8B(Ours) | 74.7 | 45.4 | 41.4 |
> | REAL-Qwen3-VL-8B + 50% random pivot | 50.0 | 43.5 | 39.3 |
> | REAL-Qwen3-VL-8B + Oracle conflict-pivots | 83.2 | 46.2 | 42.2 |
> | REAL-Qwen3-VL-8B + Oracle reasoning-pivots | 100.0 | 46.7 | 42.6 |
>
> **Q3: RPGD and Text-Dominant Reasoning.**
>
> **A3**: We appreciate this thoughtful concern. We use top-5 highly relevant and correct paragraphs from oracle Wikipedia articles for evaluation. We test three input settings: image only, text only, and standard image-text input. Results show text plays a more critical role than vision, but **joint multimodal reasoning is particularly essential for the KIVQA task**. Meanwhile, incorporating RPGD maintains comparable accuracy, indicating **it does not impair correct reasoning**.
>
> | Model| Input | E-VQA SH | E-VQA All | InfoSeek UQ | InfoSeek  UE | InfoSeek  All |
> | -| - | -| -| -| -| -|
> | Qwen3-VL-2B  | image| 16.0| 15.6 | 14.6 | 13.3| 14.0 |
> | Qwen3-VL-2B| oracle text | 40.0 | 35.9 | 38.8 | 36.9 | 37.9 |
> | Qwen3-VL-2B | image + oracle text | 54.3 | 49.7 | 57.2 | 56.7 | 56.9|
> | **REAL-Qwen3-VL-2B(Ours)** | **image + oracle text** | **54.2** | **49.4** | **57.2** | **56.6** | **56.9** |
>
> **Q4: Self-consistency Decoding.**
>
> **A4**: Thank you for your suggestions. We compare REAL with the self-consistency decoding strategy. This method yields only marginal gains because it does not fundamentally address knowledge conflicts. Even with multiple generated samples, most predictions remain unchanged due to fixed conflicting biases, **resulting in consistently inferior performance compared to RPGD**. Experimental results are shown in the table below:
>
> | Model | Time | E-VQA SH | E-VQA All | InfoSeek UQ | InfoSeek  UE | InfoSeek  All |
> | -| - | - | - | - | - | -|
> | Qwen3-VL-8B| 1× | 42.4 | 38.1| 38.0 | 39.4 | 38.7|
> | + SC (N=5)| 5×| 42.4| 38.1| 38.2 | 39.4| 38.8 |
> | + SC (N=10)| 10× | 42.7  | 38.2   | 38.4  | 39.5  | 39.0 |
> | **+ RPGD (Ours)** | **1.26×** | **45.5** | **41.4**  | **43.1**   | **45.1**     | **44.1** |
>
> 1× refers to the decoding time of vanilla Qwen3‑VL‑8B. SC (self‑consistency) samples N responses via temperature sampling and selects the most frequent answer as the final prediction.
>
> **Q5: Minor Discussion.**
>
> **A5**: Thank you for this insightful comment. Identifying correct knowledge is indeed the ultimate goal. However, retrieved evidence can be highly misleading due to retrieval biases and potential knowledge perturbations. Even for human experts and powerful LLMs such as Qwen3-VL-235B-A22B, distinguishing true knowledge from highly similar, conflicting ones is extremely difficult without external search. **Thus, conflict discrimination currently serves as a valuable intermediate objective that offers clear and measurable signals**. We have evaluated both conflict detection and right or helpful knowledge detection on the REAL-VQA test set using human experts and Qwen3-VL-235B-A22B respectively, **which verifies that directly identifying correct knowledge remains a highly challenging task**.
>
> | Detection Method | Conflict Detection (Acc) | Truth Detection (Acc) |
> | - | - | - |
> | Human Expert(w/o search) | 92.0  | 64.5  |
> | Qwen3-VL-235B-A22B       | 83.2 | 56.4  |

---

> > ### Author Rebuttal · Reviewer_ZFNr · 2026-04-03
> >
> > Thanks for the detailed response and additional discussions. Most of my concerns have been addressed, and I currently have no more questions. I will maintain my positive score.

---

> > > ### Author Response · Authors · 2026-04-04
> > >
> > > We sincerely appreciate your careful review and positive feedback. Your insightful comments have greatly helped us address issues related to knowledge conflict, and we will integrate these revisions into the next version.

---

### Decision · Program_Chairs · 2026-04-30

**Decision:**

Accept (regular)

**Comment:**

The reviewers found this paper to be technically solid and generally well executed, and most agreed that the problem of resolving conflicting retrieved knowledge in KI-VQA is important and worth studying. The rebuttal was helpful and addressed a number of practical concerns, including questions about parameter efficiency, robustness, ablations, and the contribution of the decoding strategy, which improved confidence in the empirical results. At the same time, I do not see this as a clear-cut accept. Some reviewers remained unconvinced about the conceptual novelty of the reasoning-pivot formulation, and the intuition behind the patch-shuffle-based decoding path is still not fully satisfying beyond the empirical evidence. Overall, however, the paper has enough technical merit, experimental support, and potential value to the community to warrant a weak accept, though it would benefit from a clearer positioning of its novelty and a stronger explanation of the method’s underlying rationale in the final version.